# Optimal policy for attention-modulated decisions explains human fixation behavior

Anthony I Jang[1], Ravi Sharma[2], Jan Drugowitsch[1]*

[1]Department of Neurobiology, Harvard Medical School, Boston, United States; [2]Division of Biostatistics and Bioinformatics, Department of Family Medicine and Public Health, UC San Diego School of Medicine, La Jolla, United States

**Abstract** Traditional accumulation-to-bound decision-making models assume that all choice options are processed with equal attention. In real life decisions, however, humans alternate their visual fixation between individual items to efficiently gather relevant information (Yang et al., 2016). These fixations also causally affect one's choices, biasing them toward the longer-fixated item (Krajbich et al., 2010). We derive a normative decision-making model in which attention enhances the reliability of information, consistent with neurophysiological findings (Cohen and Maunsell, 2009). Furthermore, our model actively controls fixation changes to optimize information gathering. We show that the optimal model reproduces fixation-related choice biases seen in humans and provides a Bayesian computational rationale for this phenomenon. This insight led to additional predictions that we could confirm in human data. Finally, by varying the relative cognitive advantage conferred by attention, we show that decision performance is benefited by a balanced spread of resources between the attended and unattended items.

**\*For correspondence:**
jan_drugowitsch@hms.harvard.edu

**Competing interests:** The authors declare that no competing interests exist.

## Introduction

Would you rather have a donut or an apple as a mid-afternoon snack? If we instantaneously knew their associated rewards, we could immediately choose the higher-rewarding option. However, such decisions usually take time and are variable, suggesting that they arise from a neural computation that extends over time (*Rangel and Hare, 2010*; *Shadlen and Shohamy, 2016*). In the past, such behavior has been modeled descriptively with accumulation-to-bound models that continuously accumulate noisy evidence from each choice option, until a decision boundary is reached in favor of a single option over its alternatives. Such models have been successful at describing accuracy and response time data from human decision makers performing in both perceptual and value-based decision tasks (*Ratcliff and McKoon, 2008*; *Milosavljevic et al., 2010*). Recently, we and others showed that, if we assume these computations to involve a stream of noisy samples of each item's perceptual feature (for perceptual decisions) or underlying value (for value-based decisions), then the normative strategy could be implemented as an accumulation-to-bound model (*Bogacz et al., 2006*; *Drugowitsch et al., 2012*; *Tajima et al., 2016*). Specifically, the normative strategy could be described with the diffusion decision model (*Ratcliff and McKoon, 2008*) with time-varying decision boundaries that approach each other over time.

Standard accumulation-to-bound models assume that all choice options receive equal attention during decision-making. However, the ability to drive one's attention amidst multiple, simultaneous trains of internal and external stimuli is an integral aspect of everyday life. Indeed, humans tend to alternate between fixating on different items when making decisions, suggesting that control of overt visual attention is intrinsic to the decision-making process (*Kustov and Robinson, 1996*; *Mohler and Wurtz, 1976*). Furthermore, their final choices are biased toward the item that they

looked at longer, a phenomenon referred to as a choice bias (*Shimojo et al., 2003*; *Krajbich et al., 2010*; *Krajbich and Rangel, 2011*; *Cavanagh et al., 2014*). While several prior studies have developed decision-making models that incorporate attention (*Yu et al., 2009*; *Krajbich et al., 2010*; *Towal et al., 2013*; *Cassey et al., 2013*; *Gluth et al., 2020*), our goal was to develop a normative framework that incorporates control of attention as an integral aspect of the decision-making process, such that the agent must efficiently gather information from all items while minimizing the deliberation time, akin to real life decisions. In doing so, we hoped to provide a computational rationale for why fixation-driven choice biases seen in human behavior may arise from an optimal decision strategy. For example, the choice bias has been previously replicated with a modified accumulation-to-bound model, but the model assumed that fixations are driven by brain processes that are exogenous to the computations involved in decision-making (*Krajbich et al., 2010*). This stands in contrast to studies of visual attention where fixations appear to be controlled to extract choice-relevant information in a statistically efficient manner, suggesting that fixations are driven by processes endogenous to the decision (*Yang et al., 2016*; *Hoppe and Rothkopf, 2016*; *Hayhoe and Ballard, 2005*; *Chukoskie et al., 2013*; *Corbetta and Shulman, 2002*).

We asked if the choice bias associated with fixations can be explained with a unified framework in which fixation changes and decision-making are part of the same process. To do so, we endowed normative decision-making models (*Tajima et al., 2016*) with attention that boost the amount of information one collects about each choice option, in line with neurophysiological findings (*Averbeck et al., 2006*; *Cohen and Maunsell, 2009*; *Mitchell et al., 2009*; *Wittig et al., 2018*). We furthermore assumed that this attention was overt (*Posner, 1980*; *Geisler and Cormack, 2012*), and thus reflected in the decision maker's gaze which was controlled by the decision-making process.

We first derive the complex normative decision-making strategy arising from these assumptions and characterize its properties. We then show that this strategy featured the same choice bias as observed in human decision makers: it switched attention more frequently when deciding between items with similar values, and was biased toward choosing items that were attended last, and attended longer. It furthermore led to new predictions that we could confirm in human behavior: choice biases varied based on the amount of time spent on the decision and the average desirability across both choice items. Lastly, it revealed why the observed choice biases might, in fact, be rational. Overall, our work provides a unified framework in which the optimal, attention-modulated information-seeking strategy naturally leads to biases in choice that are driven by visual fixations, as observed in human decisions.

## Results

### An attention-modulated decision-making model

Before describing our attention-modulated decision-making model, we will first briefly recap the attention-free value-based decision-making model (*Tajima et al., 2016*) that ours builds upon. This model assumes that for each decision trial, a true value associated with each item ($z_1, z_2$) is drawn from a normal prior distribution with mean $\bar{z}$ and variance $\sigma_z^2$. Therefore, $z_j \sim \mathcal{N}(\bar{z}, \sigma_z^2)$ for both $j \in \{1, 2\}$. The smaller the $\sigma_z^2$, the more information this prior provides about the true values. We assume the decision maker knows the shape of the prior, but can't directly observe the drawn true values. In other words, the decision maker a priori knows the range of values associated with the items they need to compare, but does not know what exact items to expect nor what their associated rewards will be. For example, one such draw might result in a donut and an apple, each of which has an associated value to the decision maker (i.e. satisfaction upon eating it). In each *n*th time step of length $\delta t$, they observe noisy samples centered around the true values, called *momentary evidence*, $\delta x_{j,n} | z_j \sim \mathcal{N}(z_j \delta t, 2\sigma_x^2 \delta t)$. In *Tajima et al., 2016* , the variance of the momentary evidence was $\sigma_x^2 \delta t$ rather than $2\sigma_x^2 \delta t$. We here added the factor 2 without loss of generality to relate it more closely to the attention-modulated version we introduce further below. The variance $2\sigma_x^2$ here controls how informative the momentary evidence is about the associated true value. A large $\sigma_x^2$ implies larger noise, and therefore less information provided by each of the momentary evidence samples. While the model is agnostic to the origin of these samples, they might arise from computations to infer the items' values (e.g. how much do I currently value the apple?), memory recall (e.g.

how much did I previously value the apple?), or a combination thereof (*Shadlen and Shohamy, 2016*). As the decision maker's aim is to choose the higher valued item, they ought to accumulate evidence for some time to refine their belief in the items' values. Once they have accumulated evidence for $t = N\delta t$ seconds, their posterior belief for the value associated with either item is

$$z_j | \delta x_{j,1:N} \sim \mathcal{N}\left(\frac{\sigma_x^2 \sigma_z^{-2}\bar{z} + \frac{1}{2}x_j(t)}{\sigma_x^2 \sigma_z^{-2} + \frac{1}{2}t}, \frac{\sigma_x^2}{\sigma_x^2 \sigma_z^{-2} + \frac{1}{2}t}\right), \tag{1}$$

where $x_j(t) = \sum_{n=1}^{N} \delta x_{j,n}$ is the accumulated evidence for item $j$ (*Tajima et al., 2016*). The mean of this posterior (i.e. the first fraction in brackets) is a weighted sum of the prior mean, $\bar{z}$, and the accumulated evidence, $x_j(t)$. The weights are determined by the accumulation time ($t$), and the variances of the prior ($\sigma_z^2$) and the momentary evidence ($\sigma_x^2$), which control their respective informativeness. Initially, $t = 0$ and $x_j(t) = 0$, such that the posterior mean equals that of the prior, $\bar{z}$. Over time, with increasing $t$, the influence of $x_j(t)$ becomes dominant, and the mean approaches $x_j(t)/t$ (i.e. the average momentary evidence) for a large $t$, at which point the influence of the prior becomes negligible. The posterior's variance (i.e. the second fraction in brackets) reflects the uncertainty in the decision maker's value inference. It initially equals the prior variance, $\sigma_z^2$, and drops toward zero once $t$ becomes large. In this attention-free model, uncertainty monotonically decreases identically over time for both items, reflecting the standard assumption of accumulation-to-bound models that, in each small time period, the same amount of evidence is gathered for either choice item.

To introduce attention-modulation, we assume that attention limits information about the unattended item (*Figure 1*). This is consistent with behavioral and neurophysiological findings showing that attention boosts behavioral performance (*Cohen and Maunsell, 2009*; *Cohen and Maunsell, 2010*; *Wang and Krauzlis, 2018*) and the information encoded in neural populations (*Ni et al.,*

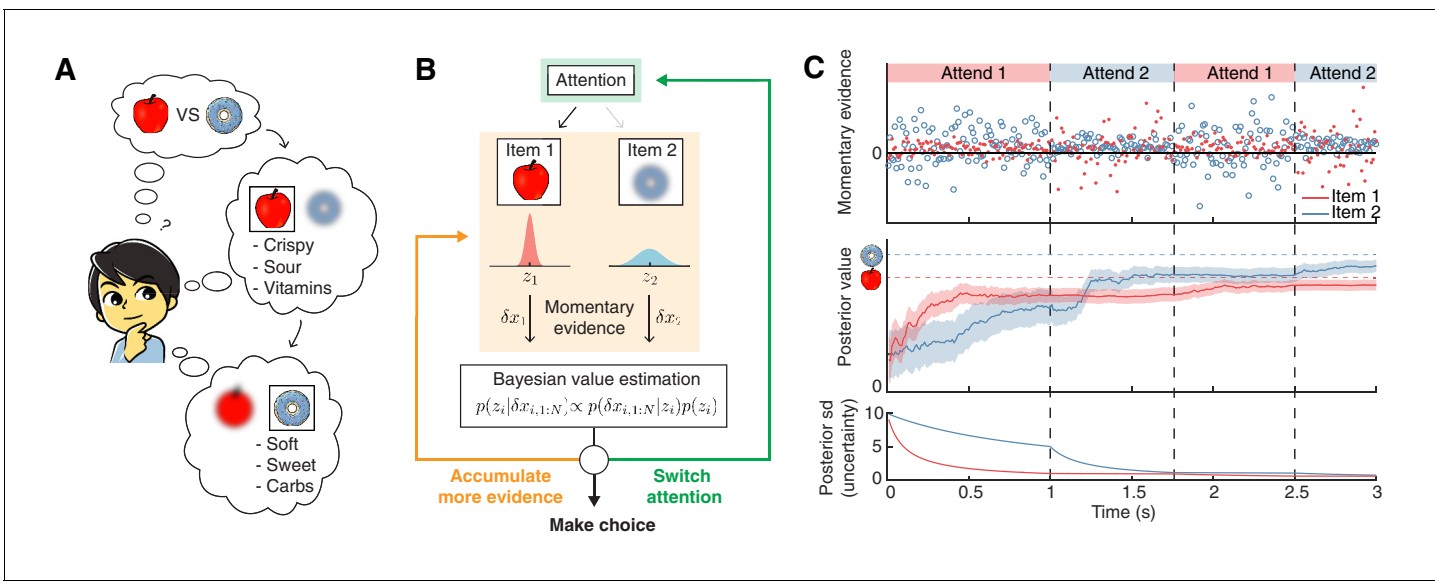

**Figure 1.** Attention-modulated evidence accumulation. (A) Schematic depicting the value-based decision-making model. When choosing between two snack items (e.g. apple versus donut), people tend to evaluate each item in turn, rather than think about all items simultaneously. While evaluating one item, they will pay less attention to the unattended item (blurred item). (B) Schematic of the value-based decision process for a single decision trial. At trial onset, the model randomly attends to one item (green box). At every time step, it accumulates momentary evidence (orange box) that provides information about the true value of each item, which is combined with the prior belief of each item's value to generate a posterior belief. Note that the momentary evidence of the attended item comes from a tighter distribution. Afterwards, the model assesses whether to accumulate more evidence (orange), make a choice (black), or switch attention to the other item (green). (C) Evolution of the evidence accumulation process. The top panel shows momentary evidence at every time point for the two items. Note that evidence for the unattended item has a wider variance. The middle panel shows how the posterior estimate of each item may evolve over time (mean ± 1SD). The horizontal dotted lines indicate the unobserved, true values of the two items. The bottom panel shows how uncertainty decreases regarding the true value of each item. As expected, uncertainty decreases faster for the currently attended item compared to the unattended one. For this descriptive figure, we used the following parameters: $z = [13, 10]$, $\sigma_x^2 = 5$, $\sigma_z^2 = 10$, $\gamma = 0.1$, $\delta t = 0.01$.

*2018*; *Ruff et al., 2018*; *Wittig et al., 2018*). To achieve this, we first assume that the total rate of evidence across both items, as controlled by $\sigma_x^2$, is fixed, and that attention modulates the relative amount of information gained about the attended versus unattended item. This 'attention bottleneck' is controlled by $\kappa$ ($0 \leq \kappa \leq 1$), such that $\kappa$ represents the proportion of total information received for the unattended item, versus $1 - \kappa$ for the attended item. The decision maker can control which item to attend to, but has no control over the value of $\kappa$, which we assume is fixed and known. To limit information, we change the momentary evidence for the attended item $j$ to $\delta x_{j,n} \sim \mathcal{N}\left(z_j \delta t, \frac{1}{1-\kappa} \sigma_x^2 \delta t\right)$, and for the unattended item $k = 3 - j$ to $\delta x_{k,n} \sim \mathcal{N}\left(z_k \delta t, \frac{1}{\kappa} \sigma_x^2 \delta t\right)$. Therefore, if $\kappa \leq \frac{1}{2}$, the variance of the unattended item increases (i.e. noisier evidence) relative to the attended item. This makes the momentary evidence less informative about $z_k$, and more informative about $z_j$, while leaving the overall amount of information unchanged (see Materials and methods). Setting $\kappa = \frac{1}{2}$ indicates equally informative momentary evidence for both items, and recovers the attention-free scenario (*Tajima et al., 2016*).

Lowering information for the unattended item impacts the value posteriors as follows. If the decision maker again accumulates evidence for some time $t = N\delta t$, their belief about item $j = 1$'s value changes from *Equation (1)* to

$$z_1|\delta x_{1,1:N} \sim \mathcal{N}\left(\frac{\sigma_x^2 \sigma_z^{-2} \bar{z} + (1-\kappa) X_1(t)}{\sigma_x^2 \sigma_z^{-2} + (1-\kappa) t_1 + \kappa t_2}, \frac{\sigma_x^2}{\sigma_x^2 \sigma_z^{-2} + (1-\kappa) t_1 + \kappa t_2}\right), \tag{2}$$

where $t_1$ and $t_2$, which sum up to the total accumulation time ($t = t_1 + t_2$), are the durations that items 1 and 2 have been attended, respectively. The accumulated evidence $X_1(t)$ now isn't simply the sum of all momentary pieces of evidence, but instead down-weights them by $\frac{1-\kappa}{\kappa}$ if the associated item is unattended (see Materials and methods). This prevents the large inattention noise from swamping the overall estimate (*Drugowitsch et al., 2014*). An analogous expression provides the posterior $z_2|\delta x_{2,1:N}$ for item 2 (see Appendix 1).

The attention modulation of information is clearly observable in the variance of the value's posterior for item 1 (*Equation 2*). For $\kappa < \frac{1}{2}$, this variance, which is proportional to the decision maker's uncertainty about the option's value, drops more quickly over time if item 1 rather than item 2 is attended (i.e. if $t_1$ rather than $t_2$ increases). Therefore, it depends on how long each of the two items have been attended to, and might differ between the two items across time (*Figure 1C*). As a result, decision performance depends on how much time is allocated to attending to each item.

The decision maker's best choice at any point in time is to choose the item with the larger expected value, as determined by the value posterior. However, the posterior by itself does not determine when it is best to stop accumulating evidence. In our previous attention-free model, we addressed the optimal stopping time by assuming that accumulating evidence comes at cost $c$ per second, and found the optimal decision policy under this assumption (*Tajima et al., 2016*). Specifically, at each time step of the decision-making process, the decision maker could choose between three possible actions. The first two actions involve immediately choosing one of the two items, which promises the associated expected rewards. The third action is to accumulate more evidence that promises more evidence, better choices, and higher expected reward, but comes at a higher cost for accumulating evidence. We found the optimal policy using dynamic programming that solves this arbitration by constructing a value function that, for each stage of the decision process, returns all expected rewards and costs from that stage onward (*Bellman, 1952*; *Bertsekas, 1995*). The associated policy could then be mechanistically implemented by an accumulation-to-bound model that accumulates the difference in expected rewards, $\Delta = \langle z_2 | \delta x_{2,1:N} \rangle - \langle z_1 | \delta x_{1,1:N} \rangle$, and triggers a choice once one of two decision boundaries, which collapse over time, is reached (*Tajima et al., 2016*).

Once we introduce attention, a fourth action becomes available: the decision maker can choose to switch attention to the currently unattended item (*Figure 1B*). If such a switch comes at no cost, then the optimal strategy would be to continuously switch attention between both items to sample them evenly across time. We avoid this physically unrealistic scenario by introducing a cost $c_s$ for switching attention. This cost may represent the physical effort of switching attention, the temporal cost of switching (*Wurtz, 2008*; *Cassey et al., 2013*), or a combination of both. Overall, this leads to a value function defined over a four-dimensional space: the expected reward difference $\Delta$, the

evidence accumulation times $t_1$ and $t_2$, and the currently attended item $y \in \{1, 2\}$ (see Appendix 1). As the last dimension can only take one of two values, we can equally use two three-dimensional value functions. This results in two associated policies that span the three-dimensional *state space* $(\Delta, t_1, t_2)$ (*Figure 2*).

## Features of the optimal policy

At any point within a decision, the model's current state is represented by a location in this 3D policy space, such that different regions in this space designate the optimal action to perform (i.e. choose, accumulate, switch). The boundaries between these regions can be visualized as contours in this 3D state space (*Figure 2A*). As previously discussed, there are two distinct policy spaces for when the decision maker is attending to item 1 versus item 2 that are symmetric to each other (*Figure 2B*).

Within a given decision, the deliberation process can be thought of as a particle that drifts and diffuses in this state space. The model starts out attending to an item at random ($y \in 1, 2$), which determines the initial policy space (*Figure 2B*). Assume an example trial where the model attends to item 1 initially ($y = 1$). At trial onset, the decision maker holds the prior belief, such that the particle starts on the origin ($\Delta = 0$, $t_1 = t_2 = 0$) which is within the 'accumulate' region. As the model accumulates evidence, the particle will move on a plane perpendicular to $t_2 = 0$, since $t_2$ remains constant while attending to item 1 (*Figure 2C*, first column). During this time, evidence about the true values of both items will be accumulated, but information regarding item 2 will be significantly noisier (as controlled by $\kappa$). Depending on the evidence accumulated regarding both items, the particle may hit the boundary for 'choose 1', 'choose 2', or 'switch (attention)'. Assume the particle hits the 'switch' boundary, indicating that the model is not confident enough to make a decision after the initial fixation to item 1. In other words, the difference in expected rewards between the two items is too small to make an immediate decision, and it is deemed advantageous to collect more information about the currently unattended item. Now, the model is attending to item 2, and the policy space switches accordingly ($y = 2$). The particle, starting from where it left off, will now move on a plane perpendicular to the $t_1$ axis (*Figure 2C*, second column). This process is repeated until the particle hits a decision boundary (*Figure 2C*, third column). Importantly, these shifts in attention are endogenously generated by the model as a part of the optimal decision strategy — it exploits its ability to control how much information it receives about either item's value.

The optimal policy space shows some notable properties. As expected, the 'switch' region in a given policy space is always encompassed in the 'accumulate' region of the other policy space, indicating that the model never switches attention or makes a decision immediately after an attention switch. Furthermore, the decision boundaries in 3D space approach each other over time, consistent with previous work that showed a collapsing 2D boundary for optimal value-based decisions without attention (*Tajima et al., 2016*). The collapsing bound reflects the model's uncertainty regarding the difficulty of the decision task (*Drugowitsch et al., 2012*). In our case, this difficulty depends on how different the true item values are, as items of very different values are easier to distinguish than those of similar value. If the difficulty is known within and across choices, the boundaries will not collapse over time, and their (fixed) distance will reflect the difficulty of the choice. However, since the difficulty of individual choices varies and is a priori unknown to the decision maker in our task, the decision boundary collapses so that the model minimizes wasting time on a choice that is potentially too difficult.

The optimal model had five free parameters that affect its behavior: (1) variance of evidence accumulation ($\sigma_x^2$), (2) variance of the prior distribution ($\sigma_z^2$), (3) cost of evidence accumulation ($c[s^{-1}]$), (4) cost of switching attention ($c_s$), and (5) relative information gain from the attended vs. unattended items ($\kappa$). The contour of the optimal policy boundaries changes in intuitive ways as these parameters are adjusted (*Figure 2—figure supplement 1*). Increasing the noisiness of evidence accumulation ($\sigma_x^2$) causes an overall shrinkage of the evidence accumulation space. This allows the model to reach a decision boundary more quickly under a relatively higher degree of uncertainty, given that evidence accumulation is less reliable but equally costly. Similarly, increasing the cost of accumulating evidence ($c$) leads to a smaller accumulation space so that the model minimizes paying a high cost for evidence accumulation. Increasing the switch cost $c_s$ leads to a smaller policy space for the 'switch' behavior, since there is an increased cost for switching attention. Similarly, decreasing the inattention noise by setting $\kappa$ closer to $\frac{1}{2}$ leads to a smaller 'switch' space because the model can

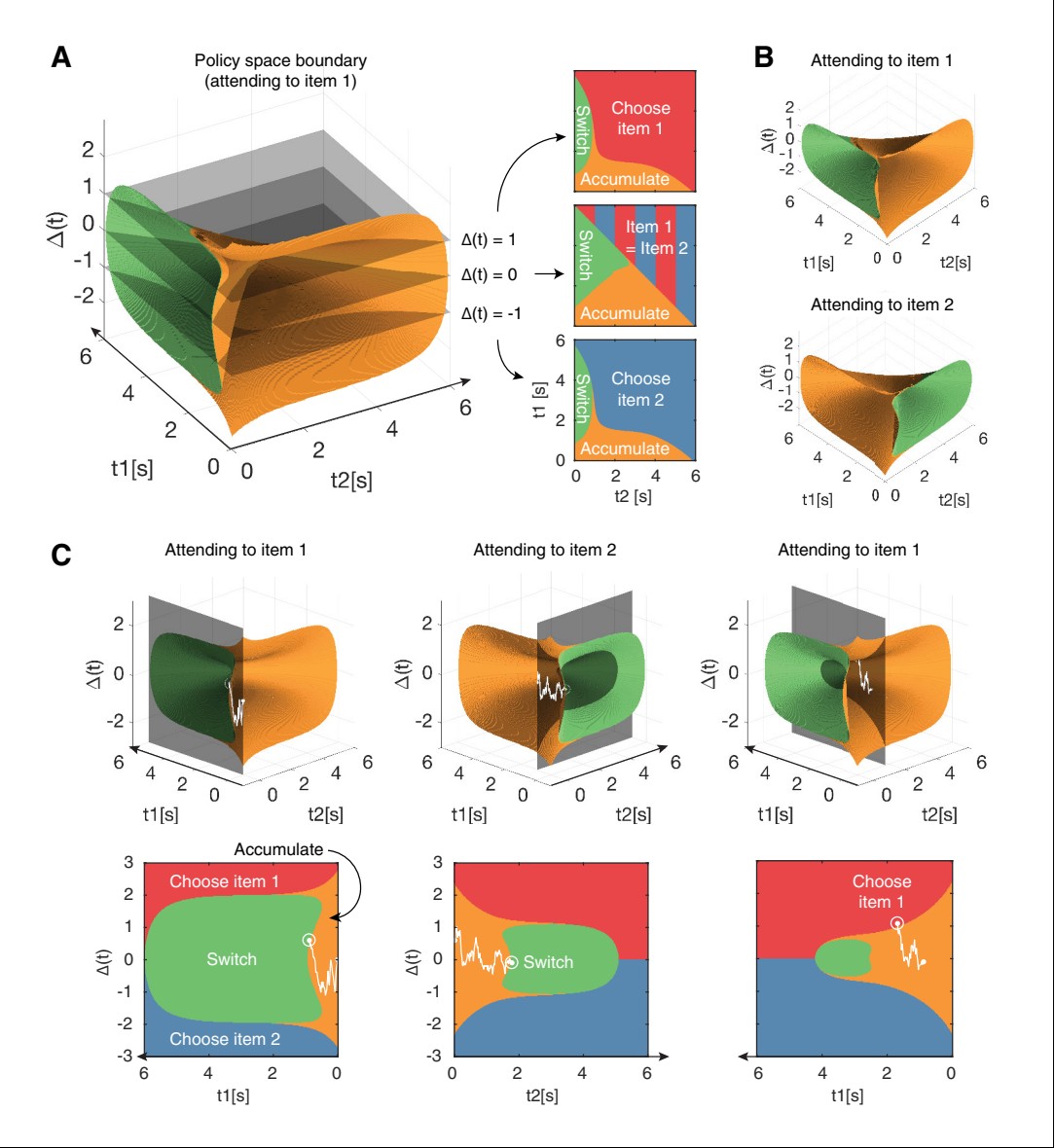

**Figure 2.** Navigating the optimal policy space. (**A**) The optimal policy space. The policy space can be divided into regions associated with different optimal actions (choose item 1 or 2, accumulate more evidence, switch attention). The boundaries between these regions can be visualized as contours in this space. The three panels on the right show cross-sections after slicing the space at different $\Delta$ values, indicated by the gray slices in the left panel. Note that when $\Delta = 0$ (middle panel), the two items have equal value and therefore there is no preference for one item over the other. (**B**) Optimal policy spaces for different values of $y$ (currently attended item). The two policy spaces are mirror-images of each other. (**C**) Example deliberation process of a single trial demonstrated by a particle that diffuses across the optimal policy space. In this example, the model starts by attending to item 1, then makes two switches in attention before eventually choosing item 1. The bottom row shows the plane in which the particle diffuses. Note that the particle diffuses on the (gray, shaded) plane perpendicular to the time axis of the unattended item, such that it only increases in $t_j$ when attending to item j. Also note that the policy space changes according to the item being attended to, as seen in (**B**). See results text for more detailed description. See *Figure 2—figure supplement 1* to view changes in the optimal policy space depending on changes to model parameters.

The online version of this article includes the following figure supplement(s) for figure 2:

**Figure supplement 1.** Changes in the optimal policy space and model behavior with adjustments in free model parameters.

obtain more reliable information from the unattended item, reducing the necessity to switch attention. To find a set of parameters that best mimic human behavior, we performed a random search over a large parameter space and selected the parameter set that best demonstrated the qualitative aspects of the behavioral data (see Appendix 1).

## The optimal policy replicates human behavior

To assess if the optimal policy features the same decision-making characteristics as human decision makers, we used it to simulate behavior in a task analogous to the value-based decision task performed by humans in *Krajbich et al., 2010*. Briefly, in this task, participants first rated their preference of different snack items on a scale of −10 to 10. Then, they were presented with pairs of different snacks after excluding the negatively rated items and instructed to choose the preferred item. While they deliberate on their choice, the participants' eye movements were tracked and the fixation duration to each item was used as a proxy for visual attention.

We simulated decision-making behavior using value distributions similar to those used in the human experiment (see Materials and methods), and found that the model behavior qualitatively reproduces essential features of human choice behavior (*Figure 3*, *Figure 3—figure supplement 1*). As expected in value-based decisions, a larger value difference among the compared items made it more likely for the model to choose the higher-valued item (*Figure 3A*; $t(38) = 105.7, p<0.001$). Furthermore, the model's mean response time (RT) decreased with increasing value difference, indicating that less time was spent on trials that were easier (*Figure 3B*; $t(38) = −11.1, p<0.001$). Of note, while human RTs appeared to drop linearly with increasing value difference, our model's drop was concave across a wide range of model parameters (*Figure 3—figure supplement 1C*). The model also switched attention less for easier trials, indicating that difficult trials required more evidence

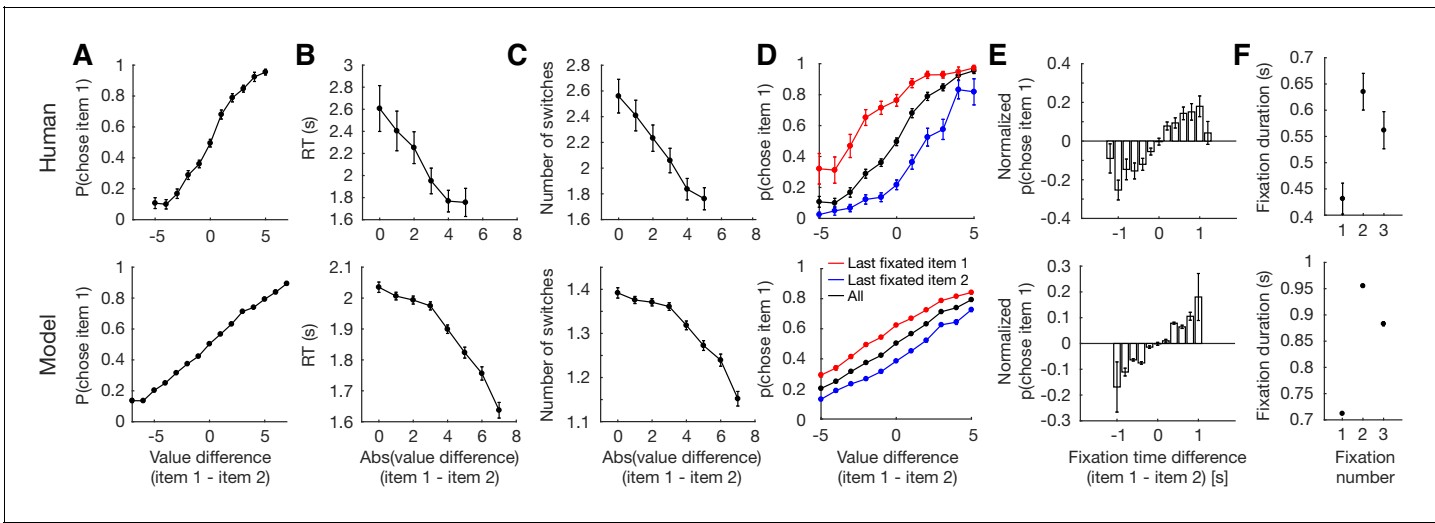

**Figure 3.** Replication of human behavior by simulated optimal model behavior (*Krajbich et al., 2010*). (A) Monotonic increase in probability of choosing item 1 as a function of the difference in value between item 1 and 2 ($t(38) = 105.7, p<0.001$). (B) Monotonic decrease in response time (RT) as a function of trial difficulty ($t(38) = −11.1, p<0.001$). RT increases with increasing difficulty. (C) Decrease in the number of attention switches as a function of trial difficulty. More switches are made for harder trials ($t(38) = −8.10, p<0.001$). (D) Effect of last fixation location on item preference. The item that was fixated on immediately prior to the decision was more likely to be chosen. (E) Attention's biasing effect on item preference. The item was more likely to be chosen if it was attended for a longer period of time ($t(38) = 5.32, p<0.001$). Since the probability of choosing item 1 depends on the degree of value difference between the two items, we normalized the p(choose item 1) by subtracting the average probability of choosing item 1 for each difference in item value. (F) Replication of fixation pattern during decision making. Both model and human data showed a fixation pattern where a short initial fixation was followed by a longer, then medium-length fixation. Error bars indicate standard error of the mean (SEM) across both human and simulated participants ($N = 39$ for both). See *Figure 3—figure supplement 2* for an analogous figure for the perceptual decision task.

The online version of this article includes the following figure supplement(s) for figure 3:

**Figure supplement 1.** Parameter-dependence of psychometric/chronometric curves, and exploration of switch rate rather than switch number for the optimal model.

**Figure supplement 2.** Replicating human perceptual decision-making behavior with the optimal model.

accumulation from both items, necessitating multiple switches in attention (*Figure 3C*; $t(38) = -8.10, p<0.001$). Since the number of switches is likely correlated with response time, we also looked at switch rate (number of switches divided by response time). Here, although human data showed no relationship between switch rate and trial difficulty, model behavior showed a positive relationship, suggesting an increased rate of switching for easier trials. However, this effect was absent when using the same number of trials as humans, and did not generalize across all model parameter values (*Figure 3—figure supplement 1D–G*).

The model also reproduced the biasing effects of fixation on preference seen in humans (*Krajbich et al., 2010*). An item was more likely to be chosen if it was the last one to be fixated on (*Figure 3D*), and if it was viewed for a longer time period (*Figure 3E*; $t(38) = 5.32, p<0.001$). Interestingly, the model also replicated a particular fixation pattern seen in humans, where a short first fixation is followed by a significantly longer second fixation, which is followed by a medium-length third fixation (*Figure 3F*). We suspect this pattern arises due to the shape of the optimal decision boundaries, where the particle is more likely to hit the 'switch' boundary in a shorter time for the first fixation, likely reflecting the fact that the model prefers to sample from both items at least once. Consistent with this, *Figure 3C* shows that the 'accumulate' space is larger for the second fixation compared to the first fixation. Of note, the attentional drift diffusion model (aDDM) that was initially proposed to explain the observed human data did not generate its own fixations, but rather used fixations sampled from the empirical distribution of human subjects. Furthermore, they were only able to achieve this fixation pattern by sampling the first fixation, which was generally shorter than the rest, separately from the remaining fixation durations (*Krajbich et al., 2010*; *Figure 4—figure supplement 3E*).

One feature that distinguishes our model from previous attention-based decision models is that attention only modulates the variance of momentary evidence without explicitly down-weighting the value of the unattended item (*Krajbich et al., 2010*; *Song et al., 2019*). Therefore, at first glance, preference for the more-attended item is not an obvious feature since our model does not appear to boost its estimated value. However, under the assumption that decision makers start out with a zero-mean prior, Bayesian belief updating with attention modulation turns out to effectively account for a biasing effect of fixation on the subjective value of items (*Li and Ma, 2019*). For instance, consider choosing between two items with equal underlying value. Without an attention-modulated process, the model will accumulate evidence from both items simultaneously, and thus have no preference for one item over the other. However, once attention is introduced and the model attends to item 1 longer than item 2, it will have acquired more evidence about item 1's value. This will cause item 1 to have a sharper, more certain likelihood function compared to item 2 (*Figure 4A*). As posterior value estimates are formed by combining priors and likelihoods in proportion to their associated certainties, the posterior of item 1 will be less biased towards the prior than that of item 2. This leads to a higher subjective value of item 1 compared to that of item 2 even though their true underlying values are equal.

This insight leads to additional predictions for how attention-modulated choice bias should vary with certain trial parameters. For instance, the Bayesian account predicts that trials with longer response times should have a weaker choice bias than trials with shorter response times. This is because the difference in fixation times between the two items will decrease over time as the model has more opportunities to switch attention. Both the human and model behavior robustly showed this pattern (*Figure 4B*; human, $t(38) = -3.25, p = 0.0024$; model, $t(38) = -32.0, p<0.001$). Similarly, choice bias should increase for trials with higher valued items. In this case, since the evidence distribution is relatively far away from the prior distribution, the posterior distribution is 'pulled away' from the prior distribution to a greater degree for the attended versus unattended item, leading to greater choice bias. Both human and model data confirmed this behavioral pattern (*Figure 4C*; human, $t(38) = 2.95, p = 0.0054$; model, $t(38) = 11.4, p<0.001$). Since response time may be influenced by the sum of the two item values and vice versa, we repeated the above analyses using a regression model that includes both value sum and response time as independent variables (see Materials and methods). The results were largely consistent for both model (effect of RT on choice bias: $t(38) = -5.73, p<0.001$, effect of value sum: $t(38) = 7.88, p<0.001$) and human (effect of RT: $t(38) = -1.32, p = 0.20$, effect of value sum: $t(38) = 2.91, p = 0.006$) behavior.

Next, we assessed how the behavioral predictions arising from the optimal model differed from those of the original attentional drift diffusion model (aDDM) proposed by *Krajbich et al., 2010*.

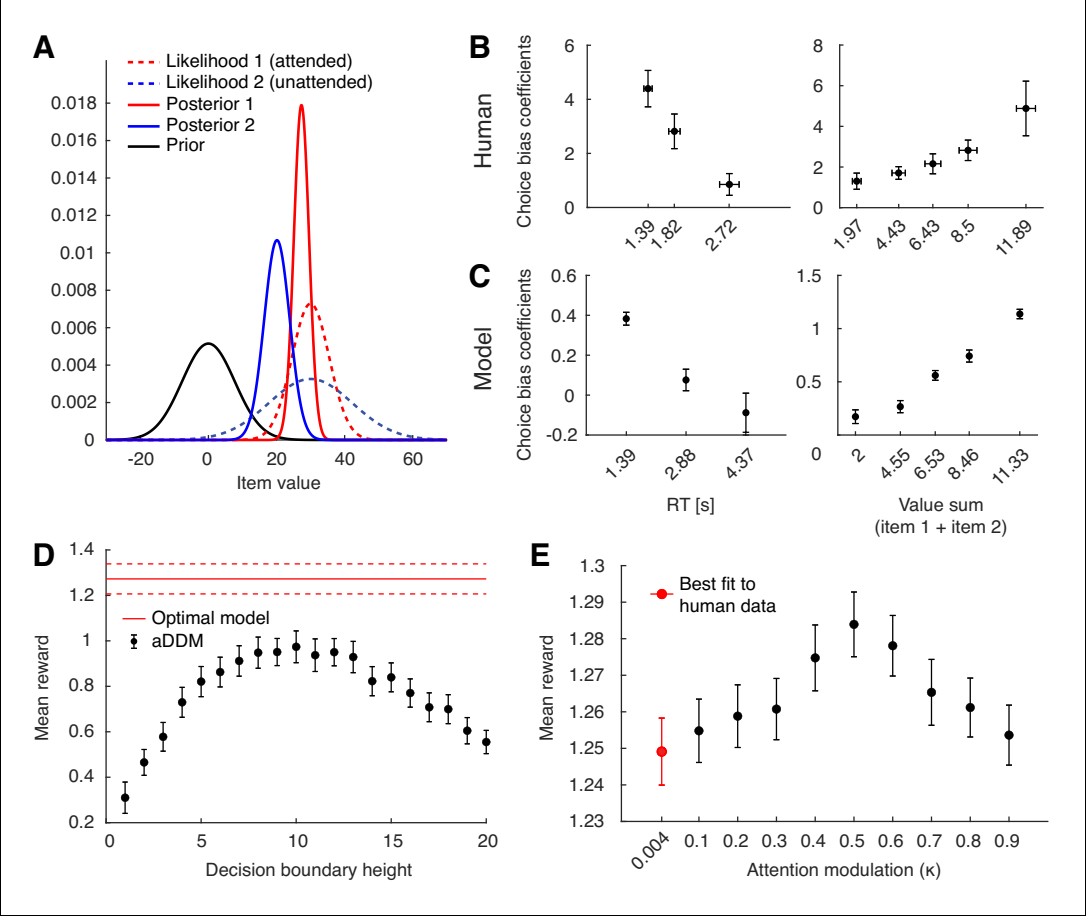

**Figure 4.** Behavioral predictions from Bayesian value estimation, and further properties of the optimal policy. (**A**) Bayesian explanation of attention-driven value preference. Attending to one of two equally-valued items for a longer time (red vs. blue) leads to a more certain (i.e. narrower) likelihood and weaker bias of its posterior towards the prior. This leads to a subjectively higher value for the longer attended item. (**B**) Effect of response time (RT; left panel; $t(38) = -3.25, p = 0.0024$) and sum of the two item values (value sum; right panel; $t(38) = 2.95, p = 0.0054$) on attention-driven choice bias in humans. This choice bias quantifies the extent to which fixations affect choices for the chosen subset of trials (see Materials and methods). (**C**) Effect of response time (left panel; $t(38) = -32.0, p<0.001$) and sum of the two item values (right panel; $t(38) = 11.4, p<0.001$) on attention-driven choice bias in the optimal model. See Materials and methods for details on how the choice bias coefficients were computed. For (**B**) and (**C**), for the left panels, the horizontal axis is binned according to the number of total fixations in a given trial. For the right panels, the horizontal axis is binned to contain the same number of trials per bin. Horizontal error bars indicate SEM across participants of the mean x-values within each bin. Vertical error bars indicate SEM across participants. (**D**) Comparing decision performance between the optimal policy and the original aDDM model. Performance of the aDDM was evaluated for different boundary heights (error bars = SEM across simulated participants). Even for the reward-maximizing aDDM boundary height, the optimal model significantly outperformed the aDDM ($t(38) = 3.01, p = 0.0027$). (**E**) Decision performance for different degrees of the attention bottleneck ($\kappa$) while leaving the overall input information unchanged (error bars = SEM across simulated participants). The performance peak at $\kappa = 0.5$ indicates that allocating similar amounts of attentional resource to both items is beneficial ($t(38) = -8.51, p<0.001$).

The online version of this article includes the following figure supplement(s) for figure 4:

**Figure supplement 1.** Effect of item values on attention switch rate and fixation duration across trials for the human data, optimal model, and aDDM.

**Figure supplement 2.** Effect of passed time on switch probability and fixation duration within trials.

**Figure supplement 3.** Additional analyses of fixation behavior and performance between human data, optimal model, and aDDM.

Unlike our model, the aDDM follows from traditional diffusion models rather than Bayesian models. It assumes that inattention to an item diminishes its value magnitude rather than increasing the noisiness of evidence accumulation. Despite this difference, the aDDM produced qualitatively similar behavioral predictions as the optimal model (see *Figure 4—figure supplement 1*, *Figure 4—figure supplement 2*, and *Figure 4—figure supplement 3* for additional behavioral comparisons between human data, the optimal model, and aDDM). We also tested to which degree the optimal model yielded a higher mean reward than the aDDM, which, despite its simpler structure, could nonetheless collect competitive amounts of reward. Given that our model provides the optimal solution to the decision problem under the current assumptions, it is expected to outperform, or at least match, the performance of alternative models. To ensure a fair comparison, we adjusted the aDDM model parameters (i.e. attentional value discounting and the noise variance) so that the momentary evidence provided to the two models has equivalent signal-to-noise ratios (see Appendix 1). Using the same parameters fit to human behavior without this adjustment in signal-to-noise ratio yielded a higher mean reward for the aDDM model ($t(76) = -14.8, p<0.001$), since the aDDM receives more value information at each time point than the optimal model. The original aDDM model fixed the decision boundaries at $\pm1$ and subsequently fit model parameters to match behavioral data. Since we were interested in comparing mean reward, we simulated model behavior using incrementally increasing decision barrier heights, looking for the height that yields the maximum mean reward (*Figure 4D*). We found that even for the best-performing decision barrier height, the signal-to-noise ratio-matched aDDM model yielded a significantly lower mean reward compared to that of the optimal model ($t(76) = 3.01, p = 0.0027$).

Recent advances in artificial intelligence used attentional bottlenecks to regulate information flow with significant associated performance gains (*Bahdanau et al., 2015*; *Gehring et al., 2017*; *Mnih et al., 2014*; *Ba et al., 2015*; *Sorokin et al., 2015*). Analogously, attentional bottlenecks might also be beneficial for value-based decision-making. To test this, we asked if paying relatively full attention on a single item at a time confers any advantages over the ability to pay relatively less reliable, but equal attention to multiple options in parallel. To do so, we varied the amount of momentary evidence provided about both the attended and unattended items while keeping the overall amount of evidence, as controlled by $\sigma_x^2$, fixed. This was accomplished by varying the $\kappa$ term. The effect of $\kappa$ on the optimal policy was symmetric around $\kappa = 0.5$, such that information gained from attended item at $\kappa = 0.2$ is equal to that of the unattended item at $\kappa = 0.8$. Setting $\kappa = 0.5$ resulted in equal momentary evidence about both items, such that switching attention had no effect on the evidence collected about either item. When tuning model parameters to best match human behavior, we found a low $\kappa \approx 0.004$, suggesting that humans tend to allocate the majority of their presumably fixed cognitive resources to the attended item. This allows for reliable evidence accumulation for the attended item, but is more likely to necessitate frequent switching of attention.

To investigate whether widening this attention bottleneck leads to changes in decision performance, we simulated model behavior for different values of $\kappa$ (0.1 to 0.9, in 0.1 increments). Interestingly, we found that mean reward from the simulated trials is greatest at $\kappa = 0.5$ and decreases for more extreme values of $\kappa$, suggesting that a more even distribution of attentional resources between the two items is beneficial for maximizing reward ($t(38) = -8.51, p<0.001$).

## Optimal attention-modulated policy for perceptual decisions

The impact of attention is not unique to value-based decisions. In fact, recent work showed that fixation can bias choices in a perceptual decision-making paradigm (*Tavares et al., 2017*). In their task, participants were first shown a target line with a certain orientation, then shown two lines with slightly different orientations. The goal was to choose the line with the closest orientation to the previously shown target. Consistent with results in the value-based decision task, the authors demonstrated that the longer fixated option was more likely to be chosen.

We modified our attention-based optimal policy to perform in such perceptual decisions, in which the goal was to choose the option that is the closest in some quantity to the target, rather than choosing the higher valued option. Therefore, our model can be generalized to any task that requires a binary decision based on some perceptual quality, whether that involves finding the brighter dot between two dots on a screen, or identifying which of the two lines on the screen is longer. Similar to our value-based case, the optimal policy for perceptual decisions was successful at

reproducing the attention-driven biases seen in humans in *Tavares et al., 2017*, (*Figure 3—figure supplement 2*).

## Discussion

In this work, we derive a novel normative decision-making model with an attentional bottleneck, and show that it is able to reproduce the choice and fixation patterns of human decision makers. Our model significantly extends prior attempts to incorporate attention into perceptual and value-based decision-making in several ways. First, we provide a unified framework in which fixations are endogenously generated as a core component of the normative decision-making strategy. This is consistent with previous work that showed that fixation patterns were influenced by variables relevant for the decision, such as trial difficulty or the value of each choice item (*Krajbich et al., 2010*; *Krajbich and Rangel, 2011*). However, prior models of such decisions assumed an exogenous source of fixations (*Krajbich et al., 2010*; *Krajbich and Rangel, 2011*) or generated fixations using heuristics that relied on features such as the salience or value estimates of the choice options (*Towal et al., 2013*; *Gluth et al., 2020*). Other models generated fixations under the assumption that fixation duration should depend on the expected utility or informativeness of the choice items (*Cassey et al., 2013*; *Ke et al., 2016*; *Song et al., 2019*). For example, (*Cassey et al., 2013*) assumed that the informativeness of each item differed, which means the model should attend to the less informative item longer in general. Furthermore, since their decision task involved a fixed-duration, attention switches also occurred at fixed times rather than being dynamically adjusted across time, as in our case with a free-response paradigm. A recent normative model supported a continuous change of attention across choice items, and so could not relate attention to the observed discrete fixation changes (*Hébert and Woodford, 2019*). Our work significantly builds on these prior models by identifying the exact optimal policy using dynamic programming, demonstrating that fixation patterns could reflect active information gathering through controlling an attentional bottleneck. This interpretation extends previous work on visual attention to the realm of value-based and perceptual decision-making (*Yang et al., 2016*; *Hoppe and Rothkopf, 2016*; *Hayhoe and Ballard, 2005*; *Chukoskie et al., 2013*; *Corbetta and Shulman, 2002*).

Second, our model posits that attention lowers the variance of the momentary evidence associated with the attended item, which enhances the reliability of its information (*Drugowitsch et al., 2014*). In contrast, previous models accounted for attention by down-weighting the value of the unattended item (*Krajbich et al., 2010*; *Krajbich and Rangel, 2011*; *Song et al., 2019*), where one would a priori assume fixations to bias choices. Our approach was inspired by neurophysiological findings demonstrating that visual attention selectively increases the firing rate of neurons tuned to task-relevant stimuli (*Reynolds and Chelazzi, 2004*), decreases the mean-normalized variance of individual neurons (*Mitchell et al., 2007*; *Wittig et al., 2018*), and reduces the correlated variability of neurons at the population level (*Cohen and Maunsell, 2009*; *Mitchell et al., 2009*; *Averbeck et al., 2006*). In essence, selective attention appears to boost the signal-to-noise ratio, or the reliability of information encoded by neuronal signals rather than alter the magnitude of the value encoded by these signals. One may argue that we could have equally chosen to boost the evidence's mean while keeping its variance constant to achieve a similar boost in signal-to-noise ratio of the attended item. However, doing so would still distinguish our model from previous accumulation-to-bound models, as Bayes-optimal evidence accumulation in this model variant nonetheless demands the use of at least three dimensions (see *Figure 2*), and could not be achieved in the two dimensions used by previous models. Furthermore, this change would have resulted in less intuitive equations for the value posterior (*Equation 2*).

Under this framework, we show that the optimal policy can be implemented as a four-dimensional accumulation-to-bound model where the particle drifts and diffuses according to the fixation duration to either item, the currently attended item, and the difference the in items' value estimates. This policy space is significantly more complex compared to previous attention-free normative models, which can be implemented in a two-dimensional space. Nevertheless, the attention-modulated optimal policy still featured a collapsing boundary in time consistent with the attention-free case (*Drugowitsch et al., 2012*; *Tajima et al., 2016*).

When designing our model, we took the simplest possible approach to introduce an attentional bottleneck into normative models of decision-making. Our aim was to provide a precise (i.e. without

approximations), normative explanation for how fixation changes qualitatively interact with human decisions rather than quantitatively capture all details of human behavior, which is likely driven by additional heuristics and features beyond the scope of our model (*Acerbi et al., 2014*; *Drugowitsch et al., 2016*). For instance, it has been suggested that normative allocation of attention should also depend on the item values to eliminate non-contenders, which we did not incorporate as a part of our model (*Towal et al., 2013*; *Gluth et al., 2020*). Perhaps as a result of this approach, our model did not provide the best quantitative fit and was unable to capture all of the nuances of the psychometric curves from human behavior, including a seemingly linear relationship between RT and trial difficulty (*Figure 3*). As such, we expect other models using approximations to have a better quantitative fit to human data (*Krajbich et al., 2010*; *Callaway et al., 2020*). Instead, a normative understanding can provide a basis for understanding limitations and biases that emerge in human behavior. Consistent with this goal, we were able to qualitatively capture a wide range of previously observed features of human decisions (*Figure 3*), suggest a computational rationale for fixation-based choice biases (*Figure 4A*), and confirm new predictions arising from our theory (*Figure 4B–C*). In addition, our framework is compatible with recent work by *Sepulveda et al., 2020* that demonstrated that attention can bias choices toward the lower-valued option if participants are instructed to choose the less desirable item (see Appendix 1).

Due to the optimal policy's complexity (*Figure 2*), we expect the nervous system to implement it only approximately (e.g. similar to *Tajima et al., 2019* for multi-alternative decisions). Such an approximation has been recently suggested by *Callaway et al., 2020*, where they proposed a model of N-alternative choice using approaches from rational inattention to approximate optimal decision-making in the presence of an attentional bottleneck. Unlike our work, they assumed that the unattended item is completely ignored, and therefore could not investigate the effect of graded shifts of attentional resources between items (*Figure 4E*). In addition, their model did not predict a choice bias in binary choices due to a different assumption about the Bayesian prior.

In our model, we assumed the decision maker's prior belief about the item values is centered at zero. In contrast, *Callaway et al., 2020* chose a prior distribution based on the choice set, centered on the average value of only the tested items. While this is also a reasonable assumption (*Shenhav et al., 2018*), it likely contributed to their inability to demonstrate the choice bias for binary decisions. Under the assumption of our zero-mean prior, formulating the choice process through Bayesian inference revealed a simple and intuitive explanation for choice biases (*Figure 4A*) (see also *Li and Ma, 2020*). This explanation required the decision maker to a-priori believe the items' values to be lower than they actually are when choosing between appetitive options, consistent with evidence that item valuations vary inversely with the average value of recently observed items (*Khaw et al., 2017*). The zero-mean prior also predicts an opposite effect of the choice bias when deciding between aversive items, such that less-fixated items should become the preferred choice. This is exactly what has been observed in human decision makers (*Armel et al., 2008*). We justified using a zero-mean bias by pointing out that participants in the decision task were allowed to rate items as having both positive or negative valence (negative-valence items were excluded from the binary decision task). However, there is some evidence that humans also exhibit choice biases when only choosing between appetitive items (*Cavanagh et al., 2014*; *Smith and Krajbich, 2018*; *Smith and Krajbich, 2019*). Although our setup suggests a zero-mean prior is required to reproduce the choice bias, the exact features and role of the Bayesian prior in human decisions still remains an open question for future work.

We show that narrowing the attentional bottleneck by setting $\kappa$ to values closer to 0 or 1 does not boost performance of our decision-making model (*Figure 4E*). Instead, spreading a fixed cognitive reserve evenly between the attended and unattended items maximized performance. This is consistent with prior work that showed that a modified drift diffusion model with a continuously varying attention would perform optimally when attention is always equally divided (*Fudenberg et al., 2018*). However, this does not necessarily imply that equally divided attention always constitutes the normative behavior. If the decision maker has already paid more attention to one item over the other within a decision, it may be optimal to switch attention and gain more information about the unattended item rather than to proceed with equally divided attention.

Parameters fit to human behavior reveal that humans tend to allocate a large proportion of their cognitive resource toward the attended item, suggesting that the benefits of an attentional bottleneck might lie in other cognitive processes. Indeed, machine learning applied to text translation

(*Bahdanau et al., 2015*; *Gehring et al., 2017*), object recognition (*Mnih et al., 2014*; *Ba et al., 2015*), and video-game playing (*Sorokin et al., 2015*) benefits from attentional bottlenecks that allow the algorithm to focus resources on specific task subcomponents. For instance, image classification algorithms that extract only the relevant features of an image for high-resolution processing demonstrated improved performance and reduced computational cost compared to those without such attentional features (*Mnih et al., 2014*). Similarly, attentional bottlenecks that appear to limit human decision-making performance might have beneficial effects on cognitive domains outside the scope of binary value-based decisions. This is consistent with the idea that the evolutionary advantage of selective attention involves the ability to rapidly fixate on salient features in a cluttered environment, thereby limiting the amount of information that reaches upstream processing and reducing the overall computational burden (*Itti and Koch, 2001*).

An open question is whether our findings can be generalized to multi-alternative choice paradigms (*Towal et al., 2013*; *Ke et al., 2016*; *Gluth et al., 2020*; *Tajima et al., 2019*). While implementing the optimal policy for such choices may be analytically intractable, we can reasonably infer that a choice bias driven by a zero-mean prior would generalize to decisions involving more than two options. However, in a multi-alternative choice paradigm where heuristics involving value and salience of items may influence attention allocation, it is less clear whether an equally divided attention among all options would still maximize reward. We hope this will motivate future studies that investigate the role of attention in more realistic decision scenarios.

## Materials and methods

Here, we provide an outline of the framework and its results. Detailed derivations are provided in Appendix 1.

### Attention-modulated decision-making model

Before each trial, $z_1$ and $z_2$ are drawn from $z_j \sim \mathcal{N}(\bar{z}, \sigma_z^2)$. $z_1$ and $z_2$ correspond to the value of each item. In each time-step $n > 0$ of duration $\delta t$, the decision maker observes noisy samples of each $z_j$. This momentary evidence is drawn from $\delta x_{j,n} | z_j \sim \mathcal{N}(z_j \delta t, \frac{1}{1-\kappa} \sigma_x^2 \delta t)$ for the attended item $j = y_n$, and $\delta x_{k,n} | z_k \sim \mathcal{N}(z_k \delta t, \frac{1}{\kappa} \sigma_x^2 \delta t)$ for the unattended item $k \neq y_n$. We measure how informative a single momentary evidence sample is about the associated true value by computing the Fisher information it provides about this value. This Fisher information sums across independent pieces of information. This makes it an adequate measure for assessing the informativeness of momentary evidence, which we assume to be independent across time and items. Computing the Fisher information results in $(1 - \kappa) \sigma_x^{-2} \delta t$ in $\delta x_{j,n}$ about $z_j$ for the attended item, and in $\kappa \sigma_x^{-2} \delta t$ in $\delta x_{k,n}$ about $z_k$ for the unattended item. Therefore, setting $\kappa \leq \frac{1}{2}$ boosts the information of the attended, and reduces the information of the unattended item, while keeping the total information about both items at a constant $(1 - \kappa) \sigma_x^{-2} \delta t + \kappa \sigma_x^{-2} \delta t = \sigma_x^{-2} \delta t$. The posterior $z_j$ for $j \in \{1, 2\}$ after $t = N \delta t$ seconds is found by Bayes' rule, $p(z_j | \delta x_{j,1:N}, y_{1:N}) \propto p(z_j) \prod_{n=1}^{N} p(\delta x_{j,n} | z_j, y_n)$, which results in *Equation (2)*. If $y_n \in \{1, 2\}$ identifies the attended item in each time-step, the attention times in this posterior are given by $t_1 = \delta t \sum_{n=1}^{N} (2 - y_n)$ and $t_2 = \delta t \sum_{n=1}^{N} (y_n - 1)$. The attention-weighted accumulated evidence is $X_1(t) = \sum_{n=1}^{N} \left( \frac{1-\kappa}{\kappa} \right)^{y_n - 1} \delta x_{1,n}$ and $X_2(t) = \sum_{n=1}^{N} \left( \frac{1-\kappa}{\kappa} \right)^{2 - y_n} \delta x_{2,n}$, down-weighting the momentary evidence for periods when the item is unattended. Fixing $\kappa = 1/2$ recovers the attention-free case of *Tajima et al., 2016*, and the associated posterior, *Equation (1)*.

We found the optimal policy by dynamic programming (*Bellman, 1952*; *Drugowitsch et al., 2012*), which, at each point in time, chooses the action that promises the largest expected return, including all rewards and costs from that point into the future. Its central component is the value function that specifies this expected return for each value of the sufficient statistics of the task. In our task, the sufficient statistics are the two posterior means, $\langle z_j | X_j(t), t_1, t_2 \rangle$ for $j \in \{1, 2\}$, the two accumulation times, $t_1$ and $t_2$, and the currently attended item $y_n$. The decision maker can choose between four actions at any point in time. The first two are to choose one of the two items, which is expected to yield the corresponding reward, after which the trial ends. The third action is to accumulate evidence for some more time $\delta t$, which comes at cost $c \delta t$, and results in more momentary evidence and a corresponding updated posterior. The fourth is to switch attention to the other item

$3 - y_n$, which comes at cost $c_s > 0$. As the optimal action is the one that maximizes the expected return, the value for each sufficient statistic is the maximum over the expected returns associated with each action. This leads to the recursive Bellman's equation that relates values with different sufficient statistics (see Appendix 1 for details) and reveals the optimal action for each of these sufficient statistics. Due to symmetries in our task, it turns out these optimal actions only depend on the difference in posterior means $\Delta$, rather than each of the individual means (see Appendix 1). This allowed us to compute the value function and associated optimal policy in the lower-dimensional $(\Delta, t_1, t_2, y)$-space, an example of which is shown in (*Figure 2*).

The optimal policy was found numerically by backwards induction (*Tajima et al., 2016*; *Brockwell and Kadane, 2003*), which assumes that at a large enough $t = t_1 + t_2$, a decision is guaranteed and the expected return equals $\Delta$. We set this time point as $t = 6s$ based on empirical observations. From this point, we move backwards in small time steps of 0.05 s and traverse different values of $\Delta$ which was also discretized into steps of 0.05. Upon completing this exercise, we are left with a three-dimensional grid with the axes corresponding to $t_1$, $t_2$ and $\Delta$, where the value assigned to each point in space indicates the optimal decision to take for the given set of sufficient statistics. The boundaries between different optimal actions can be visualized as three-dimensional manifolds (*Figure 2*).

## Model simulations

Using the optimal policy, we simulated decisions in a task analogous to the one humans performed in *Krajbich et al., 2010*. On each simulated trial, two items with values $z_1$ and $z_2$ are presented. The model attends to one item randomly ($y \in [1, 2]$), then starts accumulating noisy evidence and adjusts its behavior across time according to the optimal policy. Since the human data had a total of 39 participants, we simulated the same number of participants ($N = 39$) for the model, but with a larger number of trials. For each simulated participant, trials consisted of all pairwise combinations of values between 0 and 7, iterated 20 times. This yielded a total of 1280 trials per simulated participant.

When computing the optimal policy, there were several free parameters that determined the shape of the decision boundaries. Those parameters included the evidence noise term ($\sigma_x^2$), spread of the prior distribution ($\sigma_z^2$), cost of accumulating evidence ($c[s^{-1}]$), cost of switching attention ($c_s$), and the relative information gain for the attended vs. unattended items ($\kappa$). In order to find a set of parameters that best mimics human behavior, we performed a random search over a large parameter space and simulated behavior using the randomly selected set of parameters (*Bergstra and Bengio, 2012*). We iterated this process for 2,000,000 sets of parameters and compared the generated behavior to that of humans (see Appendix 1). After this search process, the parameter set that best replicated human behavior consisted of $c_s = 0.0065$, $c = 0.23$, $\sigma_x^2 = 27$, $\sigma_z^2 = 18$, $\kappa = 0.004$.

## Statistical analysis

The relationship between task variables (e.g. difference in item value) and behavioral measurements (e.g. response time) were assessed by estimating the slope of the relationship for each participant. For instance, to investigate the association between response times and absolute value difference (*Figure 3B*), we fit a linear regression within each participant using the absolute value difference and response time for every trial. Statistical testing was performed using one-sample t-tests on the regression coefficients across participants. This procedure was used for statistical testing involving *Figure 3B,C,E*, and *Figure 4B,C*. To test for the effect of RT and value sum on choice bias after accounting for the other variable, we used a similar approach and used both RT and value sum as independent variables in the regression model and the choice bias coefficient as the dependent variable. To test for a significant peak effect for *Figure 4E*, we used the same procedure after subtracting 0.5 from the original $\kappa$ values and taking their absolute value. To compare performance between the optimal model and the aDDM (*Figure 4D*), we first selected the best-performing aDDM model, then performed an independent-samples t-test between the mean rewards from simulated participants from both models.

To quantify the degree of choice bias (*Figure 4B,C*), we computed a choice bias coefficient. For a given group of trials, we performed a logistic regression with fixation time difference ($t_1 - t_2$) as the independent variable and a binary-dependent variable indicating whether item 1 was chosen on each trial. After performing this regression within each participant's data, we performed a t-test of

the regression coefficients against zero. The the resulting t-statistic was used as the choice bias coefficient, as it quantified the extent to which fixations affected choice in the given subset of trials.

## Data and code availability

The human behavioral data and code are available through an open source license at https://github.com/DrugowitschLab/Optimal-policy-attention-modulated-decisions (*Jang, 2021*; copy archived at https://archive.softwareheritage.org/swh:1:rev:db4a4481aa6522d990018a34c31683698da039cb/).

## Acknowledgements

We thank Ian Krajbich for sharing the behavioral data, and members of the Drugowitsch lab, in particular Anna Kutschireiter and Emma Krause, for feedback on the manuscript. This work was supported by the National Institute of Mental Health (R01MH115554, JD) and the James S McDonnell Foundation (Scholar Award in Understanding Human Cognition, grant# 220020462, JD).

## Additional information

### Funding

| Funder | Grant reference number | Author |
|---|---|---|
| National Institute of Mental Health | R01MH115554 | Jan Drugowitsch |
| James S. McDonnell Foundation | 220020462 | Jan Drugowitsch |

The funders had no role in study design, data collection and interpretation, or the decision to submit the work for publication.

### Author contributions

Anthony I Jang, Conceptualization, Formal analysis, Methodology, Writing - original draft, Writing - review and editing; Ravi Sharma, Methodology; Jan Drugowitsch, Conceptualization, Supervision, Funding acquisition, Methodology, Writing - original draft, Writing - review and editing

### Author ORCIDs

Anthony I Jang (ID) https://orcid.org/0000-0003-3073-8228
Jan Drugowitsch (ID) https://orcid.org/0000-0002-7846-0408

### Ethics

Human subjects: Human behavioral data were obtained from previously published work from the California Institute of Technology (Krajbich et al., 2010). Caltech's Human Subjects Internal Review Board approved the experiment. Written informed consent was obtained from all participants.

### Decision letter and Author response

Decision letter https://doi.org/10.7554/eLife.63436.sa1
Author response https://doi.org/10.7554/eLife.63436.sa2

## Additional files

### Supplementary files

- Source data 1. Human behavioral data readme.
- Source data 2. Human behavioral data.
- Transparent reporting form

## Data availability

The human behavioral data and code are available through an open source license archived at https://doi.org/10.5281/zenodo.4636831 copy archived at https://archive.softwareheritage.org/swh:1:rev:db4a4481aa6522d990018a34c31683698da039cb/.

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

# Appendix 1

Here, we describe in more detail the derivations of our results, and specifics of the simulations presented in the main text. Of note, we sometimes use $x|y \sim p(y)$ to specify the conditional density $p(x|y)$. Furthermore, $\mathcal{N}(\mu, \sigma^2)$ denotes a Gaussian with mean $\mu$ and variance $\sigma^2$.

## 1 Task setup

### 1.1 Latent state prior

We assume two latent states $z_j$, $j \in \{1, 2\}$, (here, the true item values) that are before each choice trial drawn from their Gaussian prior, $z_j \sim \mathcal{N}(\bar{z}_j \sigma_z^2)$, with mean $\bar{z}_j$ and variance $\sigma_z^2$. Throughout the text, we will assume $\bar{z}_1 = \bar{z}_2$, to indicate that there is no a-priori preference of one item over the other.

### 1.2 Likelihood function of momentary evidence

The decision maker doesn't observe the latent states, but instead, in each time step of size $\delta t$, observes noisy evidence about both $z_j$'s. Let us assume that, in the $n$ th such time step, the decision maker attends to item $y_n \in \{1, 2\}$. Then, they simultaneously observe $\delta x_1$ and $\delta x_2$, distributed as

$$\delta x_{j,n}|y_n, z_j \sim \mathcal{N}\left(z_j \delta t, \left(\frac{1 - \kappa}{\kappa}\right)^{|j - y_n|} \frac{\sigma_x^2}{1 - \kappa} \delta t\right), \tag{1}$$

where we have defined the attention modulation parameter $\kappa$, bounded by $0 \le \kappa \le 1$ (we will usually assume $\kappa \le \frac{1}{2}$), and the overall likelihood variance $\sigma_x^2$. For the attended item $j = y_n$, we have $|j - y_n| = 0$, such that the the variance of the momentary evidence for this item is $\sigma_x^2 \delta t / (1 - \kappa)$. For the unattended item, for which $|j - y_n| = 1$, this variance is instead $\sigma_x^2 \delta t / \kappa$. As long as $\kappa < \frac{1}{2}$ this leads to a larger variance for the unattended item than the attended item, making the momentary evidence more informative for the attended item. In particular if we quantify this information by the Fisher information in the momentary evidence $\delta x_{j,n}$ about $z_j$, then we find this information to be $(1 - \kappa)\sigma_x^{-2}\delta t$ for the attended, and $\kappa \sigma_x^{-2}\delta t$ for the unattended item. The total Fisher information across both items is thus $\sigma_x^{-2}\delta t$, independent of $\kappa$. This shows that $\sigma_x^2$ controls the total information that the momentary evidence provides about the latent states, whereas $\kappa$ controls how much of this information is provided for the attended vs. the unattended item.

### 1.3 An alternative form for the likelihood

While the above form of the likelihood has a nice, intuitive parametrization, it is notationally cumbersome. Therefore, we will here introduce an alternative variance parametrization of this likelihood that simplifies the notation in the derivations that follow. We will use this parametrization for the rest of this Appendix.

This alternative parametrization assumes the variance of the momentary evidence of the attended item to be given by $\sigma^2 \delta t = \sigma_x^2 / (1 - \kappa)$, while that of the unattended item is given by $\gamma^{-1}\sigma^2 \delta t = \sigma_x^2 \delta t / \kappa$, where the new attention modulation parameter $\gamma$ is assumed bounded by $0 \le \gamma \le 1$. Thus, the previous parameter pair $\{\sigma_x^2, \kappa\}$ is replaced by the new pair $\{\sigma^2, \gamma\}$. A $\gamma < 1$ results in an increased variance for the unattended item, resulting in less information about the value of the unattended item. Overall, the momentary evidence likelihood is given with the alternative parametrization by

$$\delta x_{j,n}|y_n, z_j \sim \mathcal{N}\left(z_j \delta t, \frac{1}{\gamma^{|j - y_n|}} \sigma^2 \delta t\right), \tag{2}$$

This is the likelihood function that we will use for the rest of this Appendix. Any of the results can easily be mapped back to the original parametrization (as used in the main text) by

$$\sigma^2 = \frac{\sigma_x^2}{1 - \kappa}, \qquad\qquad \sigma_x^2 = (1 - \kappa)\sigma^2, \tag{3}$$

$$\gamma = \frac{\kappa}{1-\kappa}, \qquad\qquad \kappa = \frac{\gamma}{\gamma+1}. \qquad (4)$$

Note that the alternative parametrization does not preserve the separation between total information and balancing the information between the attended and unattended item. In particular, the total Fisher information is now given by $(\gamma+1)\sigma^{-2}\delta t$, which depends on both $\gamma$ and $\sigma^2$.

Below we will derive the posterior $z_j$'s, given the stream of momentary evidences $[\delta x_{1,1}, \delta x_{2,1}], [\delta x_{1,2}, \delta x_{2,2}], \ldots$, and the attention sequence $y_1, y_2, \ldots$. The mean and variance of the posterior distributions represent the decision maker's belief of the items' true values given all available evidence.

## 1.4 Costs, rewards, and the decision maker's overall aim

While the posterior estimates provide information about value, it does not tell the decision maker when to stop accumulating information, or when to switch their attention. To address these questions, we need to specify the costs and rewards associated with these behaviors. For value-based decisions, we assume that the reward for choosing item $j$ is the latent state $z_j$ (i.e. the true value) associated with the item. Furthermore, we assume that accumulating evidence comes at cost $c$ per second, or cost $c\delta t$ per time step. The decision maker can only ever attend to one item, and switching attention to the other item comes at cost $c_s$ which may be composed of a pure attention switch cost, as well as a loss of time that might introduce an additional cost. As each attention switch introduces both costs, we only consider them in combination without loss of generality.

The overall aim of the decision maker is to maximize the total expected return, which consists of the expected value of the chosen item minus the total cost of accumulating evidence and attention switches. We address this maximization problem by finding the optimal policy that, based on the observed evidence, determines when to switch attention, when to accumulate more evidence, and when to commit to a choice. We initially focus on maximizing the expected return in a single, isolated choice, and will later show that this yields qualitatively similar policies as when embedding this choice into a longer sequence of comparable choices.

## 2 Bayes-optimal evidence accumulation

### 2.1 Deriving the posterior $z_1$ and $z_2$

To find the posterior over $z_1$ after having accumulated evidence $x_{1,1:N} \equiv x_{1,1}, \ldots, x_{1,N}$ for some fixed amount of time $t = N\delta t$ while paying attention to items $y_{1:N} \equiv y_1, \ldots y_N$, we employ Bayes' rule,

$$
\begin{aligned}
p(z_1 | \delta x_{1,1:N}, y_{1:N}) \ &\propto_{z_1} p(z_1) \prod_{n=1}^{N} p(\delta x_{1,n} | z_1, y_n) \\
&= \mathcal{N}(\bar{z}_1, \sigma_z^2) \prod_{n=1}^{N} \mathcal{N}\left( z\delta t, \frac{\sigma^2}{\gamma^{|1-y_n|}}\delta t \right) \qquad (5) \\
&\propto_{z_1} \mathcal{N}\left( \frac{\bar{z}_1 \sigma^2 \sigma_z^{-2} + X_1(t)}{\sigma^2 \sigma_z^{-2} + t_1 + \gamma t_2}, \frac{\sigma^2}{\sigma^2 \sigma_z^{-2} + t_1 + \gamma t_2} \right),
\end{aligned}
$$

where we have defined $X_1(t) = \sum_{n=1}^{N} \gamma^{|1-y_n|} \delta x_{1,n}$ as the sum of all attention-weighted momentary evidence up to time t, and $t_j = t - \delta t \sum_{n=1}^{N} |j - y_n|$ as the total time that item $j$ has been attended. Note that, for time periods in which item 2 is attended to, (i.e., when $y_n = 2$), the momentary evidence is down-weighted by $\gamma$. With $\delta t \to 0$, the process becomes continuous in time, such that $X_1(t)$ becomes the integrated momentary evidence, but the above posterior still holds.

Following a similar derivation, the posterior belief about $z_2$ results in

$$p(z_2 | \delta x_{2,1:N}, y_{1:N}) = \mathcal{N}\left( \frac{\bar{z}_2 \sigma^2 \sigma_z^{-2} + X_2(t)}{\sigma^2 \sigma_z^{-2} + \gamma t_1 + t_2}, \frac{\sigma^2}{\sigma^2 \sigma_z^{-2} + \gamma t_1 + t_2} \right) \qquad (6)$$

where $X_2(t) = \sum_{n=1}^{N} \gamma^{|2-y_n|} \delta x_{2,n}$. As the decision maker acquires momentary evidence independently for both items, the two posteriors are independent of each other, that is $p(z_1, z_2 | \delta x_{1,1:N}, \delta x_{2,1:N}, y_{1:N}) = p(z_1 | \delta x_{1,1:N}, y_{1:N}) p(z_2 | \delta x_{2,1:N}, y_{1:N})$.

## 2.2 The expected reward process

At each point in time, the decision maker must decide whether it's worth accumulating more evidence versus choosing an item. To do so, they need to predict how the mean estimated reward for each option might evolve if they accumulated more evidence. In this section, we derive the stochastic process that describes this evolution for item 1. The same principles will apply for item 2.

Assume that having accumulated evidence until time $t = N\delta t$, the current expected reward for item 1 is given by $\hat{r}_1(t)$, where $\hat{r}_1(t) = \langle z_1 | \delta x_{1,1:N}, y_{1:N} \rangle$ is the mean of the above posterior, *Equation (5)*. The decision maker's prediction of how the expected reward might evolve after accumulating additional evidence for $\delta t$ is found by the marginalization,

$$p(\hat{r}_1(t+\delta t)|\hat{r}_1(t), t_1, t_2, y_{N+1})$$
$$= \iint p(\hat{r}_1(t+\delta t)|\hat{r}_1(t), \delta x_{1,N+1}, t_1, t_2, y_{N+1}) p(\delta x_{1,N+1}|z_1, y_{N+1}) p(z_1|\hat{r}_1(t), t_1, t_2) d\delta x_{1,N+1} dz_1. \tag{7}$$

As the last term in the above integral shows, $\hat{r}(t)$, $t_1$ and $t_2$ fully determine the posterior $z_1$ at time $t$. We can use this posterior to predict the value of the next momentary evidence $\delta x_{1,N+1}|z_1$. This, in turn, allows us to predict $\hat{r}_1(t+\delta t)$. As all involved densities are either deterministic or Gaussian, the resulting posterior will be Gaussian as well. Thus, rather than performing the integrals explicitly, we will find the final posterior by tracking the involved means and variances, which in turn completely determine the posterior parameters.

We first marginalize over $\delta x_{1,N+1}$, by expressing $\hat{r}_1(t+\delta t)$ in terms of $\hat{r}(t)$ and $\delta x_{1,N+1}$. To do so, we use *Equation (5)* to express $\hat{r}_1(t+\delta t)$ by

$$\hat{r}_1(t+\delta t) = \frac{\bar{z}_1 \sigma^2 \sigma_z^{-2} + X_1(t) + \gamma^{|y_{N+1}-1|} \delta x_{1,N+1}}{\sigma^2 \sigma_z^{-2} + t_1 + \gamma t_2 + \gamma^{|1-y_{N+1}|} \delta t}, \tag{8}$$

where we have used $X_1(t+\delta t) = X_1(t) + \gamma^{|y_{N+1}-1|} \delta x_{1,N+1}$.

Note that, for a given $\delta x_{1,N+1}$, $\hat{r}(t+\delta t)$ is uniquely determined by $\hat{r}(t)$. $\hat{r}(t+\delta t)$ becomes a random variable once we acknowledge that, for any $z_1$, $\delta x_{1,N+1}$ is given by *Equation (2)*, which we can write as $\delta x_{1,N+1} = z_1 \delta t + \sqrt{\sigma^2 \gamma^{-|1-y_{N+1}|} \delta t} \eta_x$, where $\eta_x \sim \mathcal{N}(0,1)$. Substituting this expression into $\hat{r}_1(t+\delta t)$, and using *Equation (5)* to re-express $X_1(t)$ as $X_1(t) = \hat{r}_1(t)(\sigma^2 \sigma_z^{-2} + t_1 + \gamma t_2) - \bar{z}_1 \sigma^2 \sigma_z^{-2}$, results in

$$\hat{r}_1(t+\delta t) = \frac{\hat{r}_1(t)(\sigma^2 \sigma_z^{-2} + t_1 + \gamma t_2) + \gamma^{|1-y_{N+1}|} z_1 \delta t + \sqrt{\sigma^2 \gamma^{|1-y_{N+1}|} \delta t} \eta_x}{\sigma^2 \sigma_z^{-2} + t_1 + \gamma t_2 + \gamma^{|1-y_{N+1}|} \delta t}. \tag{9}$$

The second marginalization over $z_1$ is found by noting the distribution of $z_1$ is given by *Equation (5)*, which can be written as

$$z_1 = \hat{r}_1(t) + \sqrt{\frac{\sigma^2}{\sigma^2 \sigma_z^{-2} + t_1 + \gamma t_2}} \eta_z, \tag{10}$$

with $\eta_z \sim \mathcal{N}(0,1)$. Substituting this $z_1$ into the above expression for $\hat{r}(t+\delta t)$ results in

$$\hat{r}_1(t+\delta t) = \hat{r}_1(t) + \frac{\sqrt{\sigma^2 \gamma^{|1-y_{N+1}|} \delta t}}{\sigma^2 \sigma_z^{-2} + t_1 + \gamma t_2 + \gamma^{|1-y_{N+1}|} \delta t} \eta_x, \tag{11}$$

where we have dropped the $\eta_z$-dependent term which had a $\delta t$ pre-factor, and thus vanishes with $\delta t \to 0$. Therefore, $\hat{r}_1(t)$ evolves as a martingale,

$$\hat{r}_1(t+\delta t)|\hat{r}_1(t), t_1, t_2, y_{N+1} \sim \mathcal{N}\left(\hat{r}_1(t), \frac{\sigma^2 \gamma^{|1-y_{n+1}|}}{(\sigma^2 \sigma_z^{-2} + t_1 + \gamma t_2 + \gamma^{|1-y_{N+1}|} \delta t)^2} \delta t\right). \tag{12}$$

Using the same approach, the expected future reward for item 2 is given by

$$\hat{r}_2(t+\delta t)|\hat{r}_2(t),t_1,t_2,y_{N+1} \sim \mathcal{N}\left(\hat{r}_2(t), \frac{\sigma^2 \gamma^{|2-y_{N+1}|}}{(\sigma^2\sigma_z^{-2}+\gamma t_1+t_2+\gamma^{|2-y_{n+1}|}\delta t)^2}\delta t\right). \tag{13}$$

## 2.3 The expected reward difference process

In a later section, we will reduce the dimensionality of the optimal policy space by using the expected reward difference rather than each of the of the expected rewards separately. To do so, we define this difference by

$$\Delta(t) = \frac{\hat{r}_1(t) - \hat{r}_2(t)}{2}. \tag{14}$$

As for $\hat{r}_1(t)$ and $\hat{r}_2(t)$, we are interested in how $\Delta(t)$ evolves over time.

To find $\Delta(t+\delta t)|\Delta(t),t_1,t_2,y_{N+1}$ we can use

$$p(\Delta(t+\delta t)|\Delta(t),t_1,t_2,y_{N+1}) = p\left(\Delta(t+\delta t) = \frac{\hat{r}_1(t+\delta t)-\hat{r}_2(t+\delta t)}{2}\Big|\Delta(t) = \frac{\hat{r}_1(t)-\hat{r}_2(t)}{2},t_1,t_2,y_{N+1}\right). \tag{15}$$

As the decision maker receives independent momentary evidence for each item, $\hat{r}_1(t)$ and $\hat{r}_2(t)$ are independent when conditioned on $t_1$, $t_2$ and $y_{1:N}$. Thus, so are their time-evolutions, $\hat{r}_1(t+\delta t)|\hat{r}_1(t),\dots$ and $\hat{r}_2(t+\delta t)|\hat{r}_2(t),\dots$. With this, we can show that

$$\Delta(t+\delta t)|\Delta(t),t_1,t_2,y_{N+1} \sim$$
$$\mathcal{N}\left(\Delta(t), \frac{\sigma^2\delta t}{4}\left(\frac{\gamma^{|1-y_{N+1}|}}{(\sigma^2\sigma_z^{-2}+t_1+\gamma t_2+\gamma^{|1-y_{N+1}|}\delta t)^2} + \frac{\gamma^{|2-y_{N+1}|}}{(\sigma^2\sigma_z^{-2}+\gamma t_1+t_2+\gamma^{|2-y_{N+1}|}\delta t)^2}\right)\right). \tag{16}$$

Unsurprisingly, $\Delta(t)$ is again a martingale.

# 3 Optimal decision policy

We find the optimal decision policy by dynamic programming (**Bellman, 1952**; **Bertsekas, 1995**). A central concept in dynamic programming is the *value function* $V(\cdot)$, which, at any point in time during a decision, returns the *expected return*, which encompasses all expected rewards and costs from that point onwards into the future when following the optimal decision policy. Bellman's equation links value functions across consecutive times, and allows finding this optimal decision policy recursively. In what follows, we first focus on Bellman's equation for single, isolated choices. After that, we show how to extend the same approach to find the optimal policy for long sequences of consecutive choices.

## 3.1 Single, isolated choice

For a single, isolated choice, accumulating evidence comes at cost $c$ per second. Switching attention comes at cost $c_s$. The expected reward for choosing item $j$ is $\hat{r}_j(t)$, and is given by the mean of *Equations (5) and (6)* for $j=1$ and $j=2$, respectively.

To find the value function, let us assume that we have accumulated evidence for some time $t=t_1+t_2$, expect rewards $\hat{r}_1(t)$ and $\hat{r}_2(t)$, and are paying attention to item $y \in \{1,2\}$. These statistics fully describe the evidence accumulation state, and thus fully parameterize the value function $V_y(\hat{r}_1,\hat{r}_2,t_1,t_2)$. Here, we use $y$ as a subscript rather than an argument to $V(\cdot)$ to indicate that $y$ can only take one of two values, $y \in \{1,2\}$. At this point, we can choose among four actions. We can either immediately choose item 1, immediately choose item 2, accumulate more evidence without switching attention, or switch attention to the other item, $3-y$. The expected return for choosing immediately is either $\hat{r}_1(t)$ or $\hat{r}_2(t)$, depending on the choice. Accumulating more evidence for some time $\delta t$ results in cost $c\delta t$, and changes in the expected rewards according to $\hat{r}_j(t+\delta t)|\hat{r}_j(t),t_1,t_2,y$, as given by *Equations (12) and (13)*. Therefore, the expected return for accumulating more evidence is given by

$$-c\delta t + \left\langle V_y(\hat{r}_1(t+\delta t),\hat{r}_2(t+\delta t),t_1+|2-y|\delta t,t_2+|1-y|\delta t)|\hat{r}_1,\hat{r}_2,t_1,t_2,y\right\rangle, \tag{17}$$

where the expectation is over the time-evolution of $\hat{r}_1$ and $\hat{r}_2$, and $t_1 + |2 - y|\delta t$ and $t_2 + |1 - y|\delta t$ ensures that only the $t_y$ associated with the currently attended item is increased by $\delta t$. Lastly, switching attention comes at cost $c_s$, but does not otherwise impact reward expectations, such that the expected return associated with this action is

$$-c_s + V_{3-y}(\hat{r}_1, \hat{r}_2, t_1, t_2),\tag{18}$$

where the use of $V_{3-y}(\cdot)$ implements that, after an attention switch, item $3 - y$ will be the attended item.

By the *Bellman, 1952* optimality principle, the best action at any point in time is the one that maximizes the expected return. Combining the expected returns associated with each possible action results in Bellman's equation

$$V_y(\hat{r}_1, \hat{r}_2, t_1, t_2) = \max \left\{ \begin{array}{c} \hat{r}_1, \hat{r}_2, \\ \langle V_y(\hat{r}_1(t+\delta t), \hat{r}_2(t+\delta t), t_1 + |2-y|\delta t, t_2 + |1-y|\delta t) | \hat{r}_1, \hat{r}_2, t_1, t_2, y \rangle - c\delta t, \\ V_{3-y}(\hat{r}_1, \hat{r}_2, t_1, t_2) - c_s \end{array} \right\}.\tag{19}$$

Solving this equation yields the optimal policy for any combination of $\hat{r}_1$, $\hat{r}_2$, $t_1$, $t_2$ and $y$ by picking the action that maximizes the associated expected return, that is, the term that maximizes the left-hand side of the above equation. The optimal decision boundaries that separate the $(\hat{r}_1, \hat{r}_2, t_1, t_2, y)$-space into regions where different actions are optimal lie at manifolds in which two actions yield the same expected return. For example, the decision boundary at which it becomes best to choose item 1 after having accumulated more evidence is the manifold at which

$$\begin{aligned} V_y(\hat{r}_1, \hat{r}_2, t_1, t_2) = \\ \hat{r}_1 = \langle V_y(\hat{r}_1(t+\delta t), \hat{r}_2(t+\delta t), t_1 + |2-y|\delta t, t_2 + |1-y|\delta t) | \hat{r}_1, \hat{r}_2, t_1, t_2, y \rangle - c\delta t. \end{aligned}\tag{20}$$

In Section 6, we describe how we found these boundaries numerically.

Formulated so far, the value function is five-dimensional, with four continuous ($\hat{r}_1$, $\hat{r}_2$, $t_1$, and $t_2$) and one discrete ($y$) dimension. It turns out that it is possible to remove one of the dimensions without changing the associated policy by focusing on the expected reward difference $\Delta(t)$, *Equation (14)*, rather than the individual expected rewards. To show this, we jump ahead and use the value function property $V_y(\hat{r}_1, \hat{r}_2, t_1, t_2) + C = V_y(\hat{r}_1 + C, \hat{r}_2 + C, t_1, t_2)$ for any scalar $C$, that we will confirm in Section 5. Next, we define the value function on expected reward differences by

$$\bar{V}_y(\Delta, t_1, t_2) = V_y(\hat{r}_1, \hat{r}_2, t_1, t_2) - \frac{\hat{r}_1 + \hat{r}_2}{2} = V_y(\Delta, -\Delta, t_1, t_2).\tag{21}$$

Applying this mapping to *Equation (19)* leads to Bellman's equation

$$\bar{V}_y(\Delta, t_1, t_2) = \max \left\{ \begin{array}{c} \Delta, -\Delta, \\ \langle \bar{V}_y(\Delta(t+\delta t), t_1 + |2-y|\delta t, t_2 + |1-y|\delta t) | \Delta, t_1, t_2, y \rangle - c\delta t, \\ \bar{V}_{3-y}(\Delta, t_1, t_2) - c_s \end{array} \right\},\tag{22}$$

which is now defined over a four-dimensional rather than a five-dimensional space while yielding the same optimal policy. This also confirms that optimal decision-making doesn't require tracking individual expected rewards, but only their difference.

## 3.2 Sequence of consecutive choices

So far, we have focused on the optimal policy for a single isolated choice. Let us now demonstrate that this policy does not qualitatively change if we move to a long sequence of consecutive choices. To do so, we assume that each choice is followed by an inter-trial interval $t_i$ after which the latent $z_1$ and $z_2$ are re-drawn from the prior, and evidence accumulation starts anew. As the expected return considers all expected future rewards, it would grow without bounds for a possibly infinite sequence of choices. Thus, rather than using the value function, we move to using the average-adjusted value function, $\tilde{V}$, which, for each passed time $\delta t$, subtracts $\rho\delta t$, where $\rho$ is the average reward rate

(*Tajima et al., 2016*). This way, the value tells us if we are performing better or worse than on average, and is thus bounded.

Introducing the reward rate as an additional time cost requires the following changes. First, the average-adjusted expected return for immediate choices becomes $\hat{r}_j(t) - t_i\rho + \tilde{V}_y(\bar{z}_1, \bar{z}_2, 0, 0)$, where $-t_i\rho$ accounts for the inter-trial interval, and $\tilde{V}_y(\bar{z}_1, \bar{z}_2, 0, 0)$ is the average-adjusted value at the beginning of the next choice, where $\hat{r}_j = \bar{z}_j$, and $t_1 = t_2 = 0$. Due to the symmetry, $\tilde{V}_y(\bar{z}_1, \bar{z}_2, 0, 0)$ will be the same for both $y = 1$ and $y = 2$, such that we do not need to specify $y$. Second, accumulating evidence for some duration $\delta t$ now comes at cost $(c + \rho)\delta t$. The expected return for switching attention remains unchanged, as we assume attention switches to be instantaneous. If attention switches take time, we would need to additionally penalize this time by $\rho$.

With these changes, Bellman's equation becomes

$$\tilde{V}_y(\hat{r}_1, \hat{r}_2, t_1, t_2) = \max\left\{\begin{array}{c} \hat{r}_1 - \rho t_i + \tilde{V}_y(\bar{z}_1, \bar{z}_2, 0, 0), \hat{r}_2 - \rho t_i + \tilde{V}_y(\bar{z}_1, \bar{z}_2, 0, 0), \\ \langle \tilde{V}_y(\hat{r}_1(t+\delta t), \hat{r}_2(t+\delta t), t_1 + |2-y|\delta t, t_2 + |1-y|\delta t)|\hat{r}_1, \hat{r}_2, t_1, t_2, y\rangle - (c+\rho)\delta t, \\ \tilde{V}_{3-y}(\hat{r}_1, \hat{r}_2, t_1, t_2) - c_s \end{array}\right\}. \quad (23)$$

The resulting average-adjusted value function is shift-invariant, that is, adding a scalar to this value function for all states does not change the underlying policy (*Tajima et al., 2016*). This property allows us to fix the average-adjusted value for one particular state, such that all other average-adjusted values are relative to this state. For mathematical convenience, we choose $\tilde{V}_y(\bar{z}_1, \bar{z}_2, 0, 0) = \rho t_i$, resulting in the new Bellman's equation

$$\tilde{V}_y(\hat{r}_1, \hat{r}_2, t_1, t_2) = \max\left\{\begin{array}{c} \hat{r}_1, \hat{r}_2, \\ \langle \tilde{V}_y(\hat{r}_1(t+\delta t), \hat{r}_2(t+\delta t), t_1 + |2-y|\delta t, t_2 + |1-y|\delta t)|\hat{r}_1, \hat{r}_2, t_1, t_2, y\rangle - (c+\rho)\delta t, \\ \tilde{V}_{3-y}(\hat{r}_1, \hat{r}_2, t_1, t_2) - c_s \end{array}\right\}. \quad (24)$$

Comparing this to Bellman's equation for single, isolated choices, *Equation (19)*, reveals an increase in the accumulation cost from $c$ to $c + \rho$. Therefore, we can find a set of task parameters for which the optimal policy for single, isolated choices will mimic that for a sequence of consecutive choices. For this reason, we will focus on single, isolate choices, as they will also capture all policy properties that we expect to see for sequences of consecutive choices.

## 3.3 Choosing the less desirable option

Recent work by *Sepulveda et al., 2020* showed that when decision makers are instructed to choose the less desirable item in a similar value-based binary decision task, fixations bias choices for the lower-valued item. Here, we show that the optimal policy also makes a similar prediction. To set the goal to choosing the less desirable option, we simply flip the signs of the expected reward associated with choosing either item from $\hat{r}_j$ to $-\hat{r}_j$ in *Equation (19)*,

$$V_y(\hat{r}_1, \hat{r}_2, t_1, t_2) = \max\left\{\begin{array}{c} -\hat{r}_1, -\hat{r}_2, \\ \langle V_y(\hat{r}_1(t+\delta t), \hat{r}_2(t+\delta t), t_1 + |2-y|\delta t, t_2 + |1-y|\delta t)|\hat{r}_1, \hat{r}_2, t_1, t_2, y\rangle - c\delta t, \\ V_{3-y}(\hat{r}_1, \hat{r}_2, t_1, t_2) - c_s \end{array}\right\}. \quad (25)$$

This sign switch makes the item with the higher value the less desirable one to choose. Otherwise, the same principles apply to computing the value function and optimal policy space.

## 4 Optimal decision policy for perceptual decisions

To apply the same principles to perceptual decision-making, we need to re-visit the interpretation of the latent states, $z_1$ and $z_2$. Those could, for example, be the brightness of two dots on a screen, and the decision maker needs to identify the brighter dot. Alternatively, they might reflect the length of two lines, and the decision maker needs to identify which of the two lines is longer. Either way, the reward is a function of $z_1$, $z_2$, and the decision maker's choice. Therefore, the expected reward for choosing either option can be computed from the posterior $z$'s, *Equations (5) and (6)*. Furthermore, these posteriors are fully determined by their means, $\hat{r}_1$, $\hat{r}_2$, and the attention times, $t_1$

and $t_2$. As a consequence, we can formulate the expected reward for choosing item $j$ by the expected reward function $R_j(\hat{r}_1, \hat{r}_2, t_1, t_2)$.

What are the consequences for this change in expected reward for the optimal policy? If we assume the attention-modulated evidence accumulation process to remain unchanged, the only change is that the expected return for choosing item $j$ changes from $\hat{r}_j$ to $R_j(\hat{r}_1, \hat{r}_2, t_1, t_2)$. Therefore, Bellman's equations changes to

$$V_y(\hat{r}_1, \hat{r}_2, t_1, t_2) = \max \left\{ \begin{array}{c} R_1(\hat{r}_1, \hat{r}_2, t_1, t_2), R_2(\hat{r}_1, \hat{r}_2, t_1, t_2), \\ \langle V_y(\hat{r}_1(t+\delta t), \hat{r}_2(t+\delta t), t_1 + |2-y|\delta t, t_2 + |1-y|\delta t)|\hat{r}_1, \hat{r}_2, t_1, t_2, y\rangle - c\delta t, \\ V_{3-y}(\hat{r}_1, \hat{r}_2, t_1, t_2) - c_s \end{array} \right\}. \quad (26)$$

The optimal policy follows from Bellman's equation as before.

The above value function can only be turned into one over expected reward differences under certain regularities of $R_1$ and $R_2$, which we will not discuss further at this point. Furthermore, for the above example, we have assumed two sources of perceptual evidence that need to be compared. Alternative tasks (e.g. the random dot motion task) might provide a single source of evidence that needs to be categorized. In this case, the formulation changes slightly (see, for example, *Drugowitsch et al., 2012*), but the principles remain unchanged.

## 5 Properties of the optimal policy

Here, we will demonstrate some interesting properties of the optimal policy, and the associated value function and decision boundaries. To do so, we re-write the value function in its non-recursive form. To do so, let us first define the switch set $\mathcal{T} = \{T_1, \ldots, T_M\}$, which determines the switch times from the current time $t$ onwards. Here, $t + T_1$ is the time of the first switch after time $t$, $t + T_1 + T_2$ is the second switch, and so on. A final decision is made at $t + \bar{T}$, where $\bar{T} = \sum_{m=1}^M T_m$, after $M - 1$ switches with associated cost $(M - 1)c_s$. As the optimal policy is the one that optimizes across choices and switch times, the associated value function can be written as

$$V_y(\hat{r}_1, \hat{r}_2, t_1, t_2) = \max_{\mathcal{T}} \langle \max\{\hat{r}_1(t+\bar{T}), \hat{r}_2(t+\bar{T})\} - c\bar{T} - (M-1)c_s|\hat{r}_1, \hat{r}_2, t_1, t_2, y\rangle, \quad (27)$$

where time expectation is over the time-evolution of $\hat{r}_1(t)$ and $\hat{r}_2(t)$, that also depends on $\mathcal{T}$. In what follows, we first derive the shift-invarance of this time-evolution, and then consider its consequences for the value function, as well as the decision boundaries.

### 5.1 Shift-invariance and symmetry of the expected reward process

Let us fix some $\mathcal{T}$, some time $t$, and assume that we are currently attending item 1, $y(t) = 1$. Then, by *Equation (12)*, $\hat{r}_1(t + \bar{T})$ can be written as

$$\hat{r}_1(t+\bar{T}) = \hat{r}_1(t) + \int_0^{T_1} \frac{\sigma}{\sigma^2 \sigma_z^{-2} + (t_1 + s_1) + \gamma t_2} dB_{1,s_1} + \int_0^{T_2} \frac{\sigma\sqrt{\gamma}}{\sigma^2 \sigma_z^{-2} + (t_1 + T_1) + \gamma(t_2 + s_2)} dB_{1,s_2} \\ + \int_0^{T_3} \frac{\sigma}{\sigma^2 \sigma_z^{-2} + (t_1 + T_1 + s_3) + \gamma(t_2 + T_2)} dB_{1,s_3} + \ldots, \quad (28)$$

where the $B_{1,s_j}$'s are white noise processes associated with item 1. This shows that, for any $\mathcal{T}$, the change in $\hat{r}_1$, that is, $\hat{r}_1(t+\bar{T}) - \hat{r}_1(t)$, is independent of $\hat{r}_1(t)$. Therefore, we can shift $\hat{r}_1(t)$ by any scalar $C$, and cause an associated shift in $\hat{r}_1(t+\bar{T})$, that is

$$p(\hat{r}(t+\bar{T}) = R + C|\hat{r}_1(t) = r + C, t_1, t_2, y) = p(\hat{r}(t+\bar{T}) = R|\hat{r}_1(t) = r, t_1, t_2, y), \quad (29)$$

As this holds for any choice of $\mathcal{T}$, it holds for all $\mathcal{T}$. A similar argument establishes this property for $\hat{r}_2$.

The above decomposition of the time-evolution of $\hat{r}_1$ furthermore reveals a symmetry between $\hat{r}_1(t+\bar{T}) - \hat{r}_1(t)$ and $\hat{r}_2(t+\bar{T}) - \hat{r}_2(t)$. In particular, the same decomposition shows that $\hat{r}_1(t+\bar{T}) - \hat{r}_1(t)$ equals $\hat{r}_2(t+\bar{T}) - \hat{r}_2(t)$ if we flip $t_1$, $t_2$ and $y(t)$. Therefore,

$$p(\hat{r}_1(t+\bar{T}) = R | \hat{r}_1(t) = r, t_1 = a, t_2 = b, y = j) = p(\hat{r}_2(t+\bar{T}) = R | \hat{r}_2(t) = r, t_1 = b, t_2 = a, y = 3-j). \quad (30)$$

## 5.2 Shift-invariance of the value function

The shift-invariance of $\hat{r}_1$ and $\hat{r}_2$ implies a shift-invariance of the value function. To see this, fix some $\mathcal{T}$ and some final choice $j$, in which case the value function according to *Equation (27)* becomes

$$V_y(\hat{r}_1, \hat{r}_2, t_1, t_2) = \langle \hat{r}_j(t+\bar{T}) | \hat{r}_1, \hat{r}_2 \rangle - c\bar{T} - (M-1)c_s, \quad (31)$$

where the expectation is implicitly conditional on $t_1$, $t_2$, $y$ and $\mathcal{T}$. Due to the shift-invariance of the time-evolution of $\hat{r}_1$ and $\hat{r}_2$, adding a scalar $C$ to both $\hat{r}_1$ and $\hat{r}_2$ increases the above expectation by the same amount, $\langle \hat{r}_j(t+\bar{T}) | \hat{r}_1, \hat{r}_2 \rangle + C$. As a consequence,

$$V_y(\hat{r}_1 + C, \hat{r}_2 + C, t_1, t_2) = V_y(\hat{r}_1, \hat{r}_2, t_1, t_2) + C. \quad (32)$$

As this holds for any choice of $\mathcal{T}$ and $j$, it also holds for the maximum over $\mathcal{T}$ and $j$, and thus for the value function in general.

A similar argument shows that the value function is increasing in both $\hat{r}_1$ and $\hat{r}_2$. To see this, fix $\mathcal{T}$ and $j$ and note that increasing either $\hat{r}_1$ or $\hat{r}_2$ causes the expectation in *Equation (31)* to either remain unchanged or to increase to $\langle \hat{r}_j(t+\bar{T}) | \hat{r}_1, \hat{r}_2 \rangle + C$. Therefore, for any non-negative $C$,

$$V_y(\hat{r}_1, \hat{r}_2, t_1, t_2) \le V_y(\hat{r}_1 + C, \hat{r}_2, t_1, t_2) \le V_y(\hat{r}_1, \hat{r}_2, t_1, t_2) + C, \quad (33)$$

$$V_y(\hat{r}_1, \hat{r}_2, t_1, t_2) \le V_y(\hat{r}_1, \hat{r}_2 + C, t_1, t_2) \le V_y(\hat{r}_1, \hat{r}_2, t_1, t_2) + C. \quad (34)$$

This again holds for any choice of $\mathcal{T}$ and $j$, such that it holds for the value function in general.

For the value function on expected reward differences, $\bar{V}_y(\Delta, t_1, t_2)$, changing both $\hat{r}_1$ and $\hat{r}_2$ by the same amount leaves $\Delta$, and therefore the associated value $\bar{V}_y(\Delta, t_1, t_2)$, unchanged. In contrast, increasing only $\hat{r}_1$ or $\hat{r}_2$ by $2C$ increases or decreases $\Delta$ by $C$. Thus, we can use $V_y(\hat{r}_1, \hat{r}_2, t_1, t_2) = \bar{V}_y(\Delta, t_1, t_2) + (\hat{r}_1 + \hat{r}_2)/2$ from *Equation (21)* and substitute it into the two above inequalities to find

$$\bar{V}_y(\Delta, t_1, t_2) - C \le \bar{V}_y(\Delta \pm C, t_1, t_2) \le \bar{V}_y(\Delta, t_1, t_2) + C, \quad (35)$$

for some non-negative $C \ge 0$. This shows that $\bar{V}_y(\Delta, t_1, t_2)$ changes sublinearly with $\Delta$. However, we cannot anymore guarantee an increase or decrease in $\bar{V}_y(\cdot)$, as an increase in $\Delta$ could arise from both an increase in $\hat{r}_1$ or a decrease in $\hat{r}_2$.

## 5.3 Symmetry of the value function

The symmetry in time-evolution across $\hat{r}_1$ and $\hat{r}_2$ results in a symmetry in the value function. To show this, let us again fix $\mathcal{T}$ and $j$, such that the value function is given by *Equation (31)*. Then, by *Equation (30)*, the expectation in the value function becomes $\langle \hat{r}_{3-j}(t+\bar{T}) | \hat{r}_2, \hat{r}_1 \rangle$ if we flip $t_1$, $t_2$, and $j$, while leaving the remaining terms of *Equation (31)* unchanged. Therefore,

$$V_y(\hat{r}_1, \hat{r}_2, t_1, t_2) = V_{3-y}(\hat{r}_2, \hat{r}_1, t_2, t_2). \quad (36)$$

For the value function on expected reward differences, a flip of $\hat{r}_1$ and $\hat{r}_2$ corresponds to a sign change of $\Delta$, such that we have

$$\bar{V}_y(\Delta, t_1, t_2) = \bar{V}_{3-y}(-\Delta, t_2, t_1). \quad (37)$$

Both cases show that we are not required to find the value function for both $y = 1$ and $y = 2$ separately, as knowing one reveals the other by the above symmetry.

## 5.4 Maximum $|V_1(\cdot) - V_2(\cdot)|$ difference

By Bellman's equation, *Equation (19)*, it is best to switch attention if the expected return of accumulating evidence equals that of switching attention, that is, if

$$V_y(\hat{r}_1, \hat{r}_2, t_1, t_2) = \langle V_y(\hat{r}_1(t+\delta t), \hat{r}_2(t+\delta t), t_1 + |2-y|\delta t, t_2 + |1-y|\delta t)|\hat{r}_1, \hat{r}_2, t_1, t_2, y \rangle - c\delta t$$
$$= V_{3-y}(\hat{r}_1, \hat{r}_2, t_1, t_2) - c_s. \tag{38}$$

Before that, $V_{3-y}(\hat{r}_1, \hat{r}_2, t_1, t_2) < V_y(\hat{r}_1, \hat{r}_2, t_1, t_2) + c_s$, as otherwise, an attention switch would have already occurred. When it does, we have $V_{3-y}(\hat{r}_1, \hat{r}_2, t_1, t_2) = V_y(\hat{r}_1, \hat{r}_2, t_1, t_2) + c_s$. That is, the attention switch happens if the value of doing so exceeds that for accumulating evidence by the switch cost $c_s$. Therefore, the difference between the value functions $V_1$ and $V_2$ can never be larger than the switch cost, that is

$$|V_1(\hat{r}_1, \hat{r}_2, t_1, t_2) - V_2(\hat{r}_1, \hat{r}_2, t_1, t_2)| \leq c_s. \tag{39}$$

Once their difference equals the switch cost, a switch occurs. It is easy to see that the same property holds for the value function on expected reward differences, leading to

$$|\bar{V}_1(\Delta, t_1, t_2) - \bar{V}_2(\Delta, t_1, t_2)| \leq c_s. \tag{40}$$

## 5.5 The decision boundaries are parallel to the diagonal $\hat{r}_1 = \hat{r}_2$

Following the optimal policy, the decision maker accumulates evidence until $V_y(\hat{r}_1, \hat{r}_2, t_1, t_2) = \max\{\hat{r}_1, \hat{r}_2\}$. For all times before that, $V_y(\hat{r}_1, \hat{r}_2, t_1, t_2) > \max\{\hat{r}_1, \hat{r}_2\}$, as otherwise, a decision is made. Let us first find an expression for the decision boundaries, and then show that these boundaries are parallel to $\hat{r}_1 = \hat{r}_2$. To do so, we will in most of this section fix $t_1$, $t_2$ and $y$, and drop them for notational convenience, that is $V(\hat{r}_1, \hat{r}_2) \equiv V_y(\hat{r}_1, \hat{r}_2, t_1, t_2)$.

First, let us assume $\hat{r}_1 > \hat{r}_2$, such that $\max\{\hat{r}_1, \hat{r}_2\} = \hat{r}_1$, and item 1 would be chosen if an immediate choice is required. Therefore $V(\hat{r}_1, \hat{r}_2) \geq \hat{r}_1$ always, and $V(\hat{r}_1, \hat{r}_2) = \hat{r}_1$ once a decision is made. For a fixed $\hat{r}_1$, the value function is increasing in $\hat{r}_2$, such that reducing $\hat{r}_2$ if $V(\hat{r}_1, \hat{r}_2) > \hat{r}_1$ will at some point lead to $V(\hat{r}_1, \hat{r}_2) = \hat{r}_1$. The optimal decision boundary is the largest $\hat{r}_2$ for which this occurs. Expressed as a function of $\hat{r}_1$, this boundary on $\hat{r}_2$ is thus given by

$$\theta_{1y}(\hat{r}_1, t_1, t_2) = \max\{\hat{r}_2 \leq \hat{r}_1 : V_y(\hat{r}_1, \hat{r}_2, t_1, t_2) = \hat{r}_1\} \tag{41}$$

A similar argument leads to the optimal decision boundary for item 2. In this case, we assume $\hat{r}_2 > \hat{r}_1$, such that $V(\hat{r}_1, \hat{r}_2) \geq \hat{r}_2$ always, and $V(\hat{r}_1, \hat{r}_2) = \hat{r}_2$ once a decision is made. The sublinear growth of the value function in both $\hat{r}_1$ and $\hat{r}_2$ implies that $V(\hat{r}_1, \hat{r}_2)$ grows at most as fast as $\hat{r}_2$, such that there will be some $\hat{r}_2$ at which $V(\hat{r}_1, \hat{r}_2) > \hat{r}_2$ turns into $V(\hat{r}_1, \hat{r}_2) = \hat{r}_2$. The optimal decision boundary is the smallest $\hat{r}_2$ for which this occurs, that is

$$\theta_{2y}(\hat{r}_1, t_1, t_2) = \min\{\hat{r}_2 \geq \hat{r}_1 : V_y(\hat{r}_1, \hat{r}_2, t_1, t_2) = \hat{r}_2\} \tag{42}$$

Note that both boundaries are on $\hat{r}_2$ as a function of $\hat{r}_1$, $t_1$, $t_2$, and $y$.

To show that these boundaries are parallel to the diagonal, we will use the shift-invariance of the value function, leading, for some scalar $C$, to

$$\begin{aligned}
\theta_{1y}(\hat{r}_1, t_1, t_2) + C &= \max\{\hat{r}_2 + C \leq \hat{r}_1 + C : V_y(\hat{r}_1, \hat{r}_2, t_1, t_2) = \hat{r}_1\} \\
&= \max\{\tilde{r}_2 \leq \tilde{r}_1 : V_y(\tilde{r}_1 - C, \tilde{r}_2 - C, t_1, t_2) = \tilde{r}_1 - C\} \\
&= \max\{\tilde{r}_2 \leq \tilde{r}_1 : V_y(\tilde{r}_1, \tilde{r}_2, t_1, t_2) = \tilde{r}_1\} \\
&= \theta_{1y}(\tilde{r}_1, t_1, t_2) \\
&= \theta_{1y}(\hat{r}_1 + C, t_1, t_2),
\end{aligned} \tag{43}$$

where we have used $\tilde{r}_j = \hat{r}_j + C$. This shows that increasing $\hat{r}_1$ by some scalar $C$ shifts the boundary on $\hat{r}_2$ by the same amount. Therefore, the decision boundary for choosing item 1 is parallel to $\hat{r}_1 = \hat{r}_2$. An analogous argument for $\theta_{2y}(\cdot)$ results in

$$\theta_{2y}(\hat{r}_1, t_1, t_2) + C = \theta_{2y}(\hat{r}_1 + C, t_1, t_2), \tag{44}$$

which showing that the same property holds for the decision boundary for choosing item 2. Overall, this confirms that the decision boundaries only depend on the expected reward difference (i.e., the direction orthogonal to $\hat{r}_1 = \hat{r}_2$), confirming that it is sufficient to compute $\bar{V}(\cdot)$ instead of $V(\cdot)$.

## 5.6 Impact of re-scaled costs, rewards, and standard deviations

To investigate the impact of re-scaling all reward and cost-dependent parameters, $c$, $c_s$, $\sigma$, and $\sigma_z$, by a constant factor $\alpha$, we first show that this re-scaling causes an equal re-scaling of the reward expectation process. To do so, note that $\sigma \to \alpha\sigma$ and $\sigma_z \to \alpha\sigma_z$ causes the expected reward expectation decomposition, *Equation (28)* to yield

$$\alpha\hat{r}_1(t+\bar{T}) = \alpha\hat{r}_1(t) + \int_0^{T_1} \frac{\alpha\sigma}{\sigma^2\sigma_z^{-2} + (t_1+s_1) + \gamma t_2} dB_{1,s_1} + \int_0^{T_2} \frac{\alpha\sigma\sqrt{\gamma}}{\sigma^2\sigma_z^{-2} + (t_1+T_1) + \gamma(t_2+s_2)} dB_{1,s_2}$$
$$+ \int_0^{T_3} \frac{\alpha\sigma}{\sigma^2\sigma_z^{-2} + (t_1+T_1+s_3) + \gamma(t_2+T_2)} dB_{1,s_3} + \ldots. \tag{45}$$

That is, the expected reward process now describes the evolution of a re-scaled version, $\hat{r}_1 \to \alpha\hat{r}_1$, of the expected reward. Therefore, with slight abuse of notation, for a fixed switch set $\tau$ and final choice $j$,

$$\langle \hat{r}_j(t+T) | \alpha\hat{r}_1, \alpha\hat{r}_2; \alpha\sigma, \alpha\sigma_z \rangle = \alpha\langle \hat{r}_j(t+T) | \hat{r}_1, \hat{r}_2; \sigma, \sigma_z \rangle, \tag{46}$$

where we have made explicit the dependency on $\sigma$ and $\sigma_z$.

To show the effect of this on the value function, keep again $\tau$ and $j$ fixed, and use $c \to \alpha c$ and $c_s \to \alpha c_s$, resulting in the value function

$$\begin{aligned} V_y(\alpha\hat{r}_1, \alpha\hat{r}_2, t_1, t_2; \alpha c, \alpha c_s, \alpha\sigma, \alpha\sigma_z) &= <\hat{r}_j | \alpha\hat{r}_1, \alpha\hat{r}_2; \alpha\sigma, \alpha\sigma_z> - \alpha c\bar{T} - \alpha(M-1)c_s \\ &= \alpha\left( <\hat{r}_j | \hat{r}_1, \hat{r}_2; \sigma, \sigma_z> - c\bar{T} - (M-1)c_s \right), \end{aligned} \tag{47}$$

which establishes that

$$V_y(\alpha\hat{r}_1, \alpha\hat{r}_2, t_1, t_2; \alpha c, \alpha c_s, \alpha\sigma, \alpha\sigma_z) = \alpha V_y(\hat{r}_1, \hat{r}_2, t_1, t_2; c, c_s, \sigma, \sigma_z) \tag{48}$$

As this holds for all $\tau$ and $j$, it is true in general. Therefore, re-scaling all costs, rewards, and standard deviations of prior and likelihood results in equivalent re-scaling of the value function, and an analogous shift of switch and decision boundaries.

# 6 Simulation details

## 6.1 Computing the optimal policy

In Section 3, we described the Bellman equation (*Equation (22)*) which outputs the expected return given these four parameters: currently attended item ($y$), reward difference ($\Delta$), expected return for accumulating more evidence, and expected return for switching attention. Note that the symmetry of the value function (Section 5) allows us to drop $-\Delta$ from the original *Equation (22)*. Solving this Bellman equation provides us with a four-dimensional 'policy space' which assigns the optimal action to take at any point in this space defined by the four parameters above.

The solution to the optimal policy can be found numerically by backwards induction (*Tajima et al., 2016*). To do so, first we assume some large $t = t_1 + t_2$, where a decision is guaranteed. In this case, $V_y(\Delta, t_1, t_2) = \max\{-\Delta, \Delta\} = |\Delta|$ for both $y = 1$ and $y = 2$. We call this the base case. From this base case, we can move one time step backwards in $t_1$ ($y = 1$):

$$\bar{V}_1(\Delta, t_1 - \delta t, t_2) = \max\left\{ \begin{array}{c} \Delta, \\ \langle \bar{V}_1(\Delta, t_1, t_2) | \Delta, t_1, t_2 \rangle - c\delta t, \\ \bar{V}_2(\Delta, t_1 - \delta t, t_2) - c_s \end{array} \right\}, \tag{49}$$

The second expression in the maximum can be evaluated, since we assume a decision is made at

time t. But $\bar{V}_2(\Delta, t_1 - \delta t, t_2) - c_s$, which is the value function for switching attention, is unknown. This unknown value function is given by

$$\bar{V}_2(\Delta, t_1 - \delta t, t_2) = \max \left\{ \begin{array}{c} \Delta, \\ \langle \bar{V}_2(\Delta, t_1 - \delta t, t_2 + \delta t) | \Delta, t_1, t_2 \rangle - c\delta t, \\ \bar{V}_1(\Delta, t_1 - \delta t, t_2) - c_s \end{array} \right\}, \tag{50}$$

In this expression, the second term can again be found, but $\bar{V}_1(\Delta, t_1 - \delta t, t_2) - c_s$ is unknown. Looking at the two expressions above, we see that under the parameters $(\Delta, t_1 - \delta t, t_2)$, $V_1 \geq V_2 - c_s$, and $V_2 \geq V_1 - c_s$, which cannot both be true. Therefore, we first assume that $V_1$ is not determined by $V_2 - c_s$, removing the $V_2 - c_s$ term from the maximum. This allows us to find $\bar{V}_1(\Delta, t_1 - \delta t, t_2)$ in *Equation (49)*. Then, we compute *Equation (50)* including the $V_1 - c_s$ term. If we find that $V_2 = V_1 - c_s$, then $V_1 \neq V_2 - c_s$, which means the $V_2 - c_s$ term could not have mattered in *Equation (49)*, and we are done. If not, we re-compute $V_1$ with the $V_2 - c_s$ term included, and we are done. Therefore, we were able to compute $V_1$ and $V_2$ under the parameters $(\Delta, t_1 - \delta t, t_2)$ using information about $\bar{V}_1(\Delta, t_1, t_2)$ and $\bar{V}_2(\Delta, t_1 - \delta t, t_2 + \delta t)$.

Using the same approach, we can find $V_{1,2}(\Delta, t_1, t_2 - \delta t)$ based on $\bar{V}_1(\Delta, t_1 - \delta t, t_2 + \delta t)$ and $\bar{V}_2(\Delta, t_1, t_2)$. Thus, given that we know $V_y(\Delta, t_1, t_2)$ above a certain $t = t_1 + t_2$, we can move backwards to compute $V_1$ and $V_2$ for $(\Delta, t_1 - \delta t, t_2)$, then $(\Delta, t_1 - 2\delta t, t_2)$, and so on, until $(\Delta, 0, t_2)$ for all relevant values of $\Delta$. Subsequently, we can do the same moving backwards in $t_2$, solving for $V_y(\Delta, t_1, t_2 - \delta t)$, $V_y(\Delta, t_1, t_2 - 2\delta t)$, ..., $V_y(\Delta, t_1, 0)$. Following this, we can continue with the same procedure from $V_y(\Delta, t_1 - \delta t, t_2 - \delta t)$, until we have found $V_{1,2}$ for all combinations of $t_1$ and $t_2$.

In practice, the parameters of the optimal policy space were discretized to allow for tractable computation. We set the large time at which decisions are guaranteed at $t = 6s$, which we determined empirically. Time was discretized into steps of $\delta t = 0.05s$. The item values, and their difference ($\Delta$) were also discretized into steps of 0.05.

Upon completing this exercise, we now have two 3-dimensional optimal policy spaces. The decision maker's location in this policy space is determined by $t_1$, $t_2$, and $\Delta$. Each point in this space is assigned an optimal action to take (choose item, accumulate more evidence, switch attention) based on which expression was largest in the maximum of the respective Bellman equation. The decision maker moves between the two policy spaces depending on which item they are attending to ($y \in [1, 2]$).

In order to find the three-dimensional boundaries that signify a change in optimal action to take, we took slices of the optimal policy space in planes of constant $\Delta$'s. We found the boundary between different optimal policies within each of these slices. We in turn approximated the three-dimensional contour of the optimal policy boundaries by collating them along the different $\Delta$'s.

## 6.2 Finding task parameters that best match human behavior

In computing the optimal policy, there were several free parameters that determined the shape of the policy boundaries, thereby affecting the behavior of the optimal model. These parameters included $\sigma^2$, $\sigma_z^2$, $c$, $c_s$, and $\gamma$. Our goal was to find a set of parameters that qualitatively mimic human behavior as best as possible. To do so, we performed a random search over the following parameter values: $c_s \in [0.001, 0.05]$ (steps size 0.001), $c \in [0.01, 0.4]$ (steps size 0.01), $\sigma^2 \in [1, 100]$ (step size 1), $\sigma_z^2 \in [1, 100]$ (step size 1), $\gamma \in [0.001, 0.01]$ (step size 0.001) (*Bergstra and Bengio, 2012*).

To find the best qualitative fit, we simulated behavior from a randomly selected set of parameter values (see next section for simulation procedure). From this simulated behavior, we evaluated the match between human and model behavior by applying the same procedure to each of *Figure 3B, C,E*. For each bin for each plot, we subtracted the mean values between the model and human data, then divided this difference by the standard deviation of the human data corresponding to that bin, essentially computing the effect size of the difference in means. We computed the sum of these effect sizes for every bin, which served as a metric for how qualitatively similar the curves were between the model and human data. We performed the same procedure for all three figures, and ranked the sum of the effect sizes for all simulations. We performed simulations for over 2,000,000 random sets of parameter values. The set of parameters for which our model best replicated human

behavior according to the above criteria was $c_s = 0.0065$, $c = 0.23$, $\sigma^2 = 27$, $\sigma_z^2 = 18$, $\gamma = 0.004$.

## 6.3 Simulating decisions with the optimal policy

The optimal policy allowed us to simulate decision making in a task analogous to the one humans performed in *Krajbich et al., 2010*. For a given set of parameters, we first computed the optimal policy. In a simulated trial, two items with values $z_1$ and $z_2$ are presented. At trial onset, the model attends to an item randomly ($y \in [1, 2]$), and starts accumulating noisy evidence centered around the true values. At every time step ($\delta t = 0.05$), the model evaluates $\Delta$ using the mean of the posteriors between the two items (see *Equations (5) and (6)*). Then, the model performs the optimal action associated with its location in the optimal policy space. If the model makes a decision, then the trial is over. If the model instead accumulates more evidence, then the above procedure is repeated for the next time step. If the model switches attention, it does not obtain further information about either item, but switches attention to the other item. Switching attention allows for more reliable evidence from the now-attended item, and also switches the optimal policy space to the appropriate one (see *Figure 2*).

To allow for a relatively fair comparison between the model and human data, we simulated the same number of subjects ($N = 39$) for the model, but with a larger number of trials. For each simulated subject, trials were created such that all pairwise combinations of values between 0 and 7 were included, and this was iterated 20 times. This yielded a total of 1280 trials per subject.

## 6.4 Attention diffusion model

In order compare the decision performance of the optimal model to that of the original attentional drift diffusion model (aDDM) proposed by *Krajbich et al., 2010*, we needed to ensure that neither model had an advantage by receiving more information. We did so by making sure that the signal-to-noise ratios of evidence accumulation of both models were identical. In aDDM, the evidence accumulation evolved according to the following process, in steps of 0.05 s (assuming y = 1):

$$v_t = v_{t-1} + d(z_1 - \gamma_k z_2) + \eta_t, \tag{51}$$

where $v_t$ is the relative decision value that represents the subjective value difference between the two items at time $t$, $d$ is a constant that controls the speed of integration (in $ms^{-1}$), $\gamma_k$ controls the biasing effect of attention, and $\eta_t \sim \mathcal{N}(0, \sigma^2)$ is a normally distributed random variable zero mean and variance $\sigma^2$. Written differently, the difference in the attention-weighted momentary evidence between item 1 and item 2 can be expressed as

$$\begin{aligned} \delta\Delta = d(z_1 - \gamma_k z_2) + \eta_t \quad &\sim \mathcal{N}\big(d(z_1 - \gamma_k z_2), \sigma^2\big) \\ &\sim \mathcal{N}\big(k(z_1 - \gamma_k z_2)\delta t, \sigma_k^2 \delta t\big), \end{aligned} \tag{52}$$

where $d$ and $\sigma^2$ were replaced by $k\delta t$, and $\sigma_k^2 \delta t$, respectively. Here, the variance term $\sigma_k^2 \delta t$ can be split into two parts, such that the $\delta\Delta$ term can be expressed as

$$\delta\Delta \sim \mathcal{N}\left(z_1 k\delta t, \frac{1}{2}\sigma_k^2 \delta t\right) - \mathcal{N}\left(\gamma_k z_2 k\delta t, \frac{1}{2}\sigma_k^2 \delta t\right). \tag{53}$$

The signal-to-noise ratios (i.e. the ratio of mean over standard deviation) of the two terms in the above equation are $\frac{z_1 k\delta t}{\sqrt{\frac{\delta t}{2}}\sigma_k}$ and $\frac{z_2 k\delta t}{\frac{1}{\gamma_k}\sigma_k \sqrt{\frac{\delta t}{2}}}$, respectively.

Continuing to assume $y = 1$, in the Bayes-optimal model, evidence accumulation evolves according to

$$\begin{aligned} \delta x_1 \quad &\sim \mathcal{N}\big(z_1 \delta t, \sigma_b^2 \delta t\big), \\ \delta x_2 \quad &\sim \mathcal{N}\big(z_2 \delta t, \gamma_b^{-1} \sigma_b^2 \delta t\big). \end{aligned} \tag{54}$$

Therefore, the difference in the attention-weighted momentary evidence between item 1 and item 2 can be expressed as:

$$\begin{aligned}
\delta\Delta \quad &\sim \mathcal{N}\left(z_1\delta t, \sigma_b^2\delta t\right) - \gamma_b \mathcal{N}\left(z_2\delta t, \gamma_b^{-1}\sigma_b^2\delta t\right) \\
&\sim \mathcal{N}\left(z_1\delta t, \sigma_b^2\delta t\right) - \mathcal{N}\left(\gamma_b z_2\delta t, \gamma_b\sigma_b^2\delta t\right).
\end{aligned} \tag{55}$$

The signal-to-noise ratios of the two terms in the above equation are $\frac{z_1\delta t}{\sqrt{\delta t}\sigma_b}$ and $\frac{z_2\delta t}{\frac{1}{\sqrt{\gamma_b}}\sigma_b\sqrt{\delta t}}$, respectively.

In order to match the signal-to-noise ratios of the two models, we set equal their corresponding expressions, to find the following relationship between the parameters of the two models:

$$\begin{aligned}
k &= 1, \\
\sigma_k^2 &= 2\sigma_b^2, \\
\gamma_k &= \sqrt{\gamma_b}.
\end{aligned} \tag{56}$$

Therefore, we simulated the aDDM with model parameters $\gamma_k = \sqrt{\gamma_b}$ and $\sigma_k^2 = 2\sigma_b^2$.

In the original aDDM model, the model parameters were estimated by fitting the model behavior to human behavior after setting a decision threshold at $\pm1$. Since we adjusted some of the aDDM parameters, we instead iterated through different decision thresholds (1 through 10, in increments of 1) and found the value that maximizes model performance. To keep it consistent with behavioral data, we generated 39 simulated participants that each completed 200 trials where the two item values were drawn from the prior distribution of the optimal policy model, $z_j \sim \mathcal{N}\left(\bar{z}, \sigma_z^2\right)$ using both the optimal model and the aDDM model.

