## [Decision Letter]

**Acceptance summary:**

Although recent research has described the interplay between attention and decision-making processes, the merits of employing selective attention during choice tasks remain largely underexplored. This paper closes this gap by offering a normative framework that specifies how reward-maximizing agents should employ attention while making value-based decisions. This framework, asides its theoretical importance, makes contact with the existing empirical literature while also providing detailed behavioural predictions that can be tested in future experiments.

**Decision letter after peer review:**

Thank you for submitting your article "Optimal policy for attention-modulated decisions explains human fixation behavior" for consideration by *eLife*. Your article has been reviewed by 3 peer reviewers, including Konstantinos Tsetsos as the Reviewing Editor and Reviewer #, and the evaluation has been overseen by Joshua Gold as the Senior Editor.

The reviewers have discussed the reviews with one another and the Reviewing Editor has drafted this decision to help you prepare a revised submission.

Summary:

In this paper the authors derive a normative model of attentional switching during binary value-based decisions. The model the authors propose assumes that attentional allocation is under the control of the agent, who on each moment decides among halting deliberation and choosing an alternative, carrying on sampling from the currently attended alternative, switching attention to the other alternative. Under certain assumptions about deliberation and attentional allocation costs, the authors derive the optimal policy, showing also that this policy reproduces key aspects of human behaviour. The approach here stands in contrast with the majority of past research on this topic, which has assumed that attentional allocation is exogenous and that it influences- but is not influenced by- the ongoing choice process. All reviewers agreed that this is an interesting and timely article that takes a normative approach to the important issue of attentional allocation in decision-making tasks. However, the reviewers also agreed that there are several points that will need to be addressed in a revision.

Essential revisions

1. Link between the normative model and human data.

One of the main claims in the paper is that the normative model reproduces key aspects of human behaviour in a value-based binary choice task reported elsewhere (Krajbich et al., 2010). This claim needs to be backed up by additional analyses and clarifications. In particular:

a. In Figure 3A, the model predicts a linear choice curve (as opposed to a sigmoidal in the human data) while, also unlike in the data, the RT curves are concave. Are the shapes of these curves a general feature of the model or do they emerge for a certain parametrisation? Would for example the results look different with fixed bounds?

b. Similarly, in Figure 3D, the model shows that the last fixation influences probability of choice equally across value differences. In the human data this effect diminishes as value difference increases. Is this prediction or a feature of the model?

c. In Figure 3C the authors focus on the number of switches, which however are highly correlated with RTs. Can the authors show in addition the number of switches per unit of time (equivalent to probability of switching) in the human data and in their model?

d. In Figure 4B-C, RTs and "value sum" may covary, with high value sums presumably resulting in shorter RTs. Does the value sum effect persist in the model and in the data once the influence of RTs is accounted for?

2. Comparison between the normative and aDDM models.

The two models differ in the way they conceptualise attentional switching (exogenous vs. endogenous). Does this difference manifest itself in the predictions the two models make? And can the human data distinguish between the two models? The authors comment in the Discussion that their model is more constrained by normative considerations, and may thus fit the data worse than other ad-hoc models. Asides quantitative fitting, which the authors may elect to perform, we would like to see a qualitative comparison between the two models.

a. How does the probability of switching changes as a function of time (within trial) and as a function of absolute value difference and sum (across trials) in the two models and in the data?

b. How does the fixation duration changes as a function of time (within trial) and as a function of absolute value difference and sum (across trials) in the two models and in the data?

c. The model seems to predict a non-negligible number of single fixation trials. How does this align with the data and the aDDM predictions?

d. How do reaction time distributions look like under the normative model? Are these comparable to the RT distributions in the data?

e. Please also show the aDDM predictions together with the novel predictions made by the normative model in Figure 4C.

f. The comparison between the two models is currently done on the basis of mean reward. How do the predictions of the models compare to the mean reward accrued by human participants in Krajbich et al? Please also clarify in the main text that the models are not compared on the basis of goodness of fit. In particular, in the paragraph starting in line 260, terms such as "outperformed", "competitive performance", "comparison" are ambiguous.

g. Please explain why the model shows the pattern that is demonstrated in Figure 3F. Is this pattern also predicted by the aDDM?

3. Comprehensibility of the modelling.

The authors have done an admirable job of boiling down some heavy mathematics (in the SI) into just the key steps in the main text. A lot of this builds on prior work in Tajima et al. Still, the authors could do more to explain the math, which would make this paper more self-contained and really help the reader understand the core ideas/intuitions.

a. Equation 1 – it isn't obvious how the mean will behave over time or where this expressions comes from. It would be helpful to state that the z part is the prior, and that as t->infinity the σ terms become negligible and the expression converges to x/t, namely the true value. The σ-squared terms seem to come out of nowhere. In the methods the authors explain that this has to do with Fisher information, but most readers won't know what that is or why it is the appropriate thing to include here. Also, σ_x_ is defined later in the text, but should be defined up front here.

b. Between Equation 1 and 2: In the means for δ_x_, one has a δ_t_ multiplying z and one doesn't. Why? Is this a typo? The δ_t_ should always be there, but again, it isn't obvious.

c. Equation 2: Why does the mean have a z1| at the beginning of it? Is that a typo?

d. Could the authors elaborate more on why/when the decision maker chooses to switch attention? They say that the decision at each time point only depends on the difference in posterior means but Figure 2c seems to indicate that if the difference in posterior means stays constant over a period of time, then the process enters the "switch" zone and shifts attention.

4. Model assumptions.

a. The model assumes that attention can take one of two discrete states. However, attention is often regarded as operating in a more graded fashion, which would necessitate parameter kappa in the model to be free to change within a trial. This will probably render the model intractable. However, a more viable and less radical assumption, would be to allow for a third "divided attention" mode in which the agent samples equally from both alternatives (divided attention mode). If this extension is technically challenging, one conceptual question that the authors can discuss in the paper, is whether the normative model would ever switch away from this divided attention mode.

b. The authors need to assume a certain prior, namely z_bar = 0, in order to always get a positive effect of attention. This seems like an important controversy in the model; it is a noticeably non-Bayesian feature of a Bayesian model. The authors try to explain this away by noting that the original rating scale included both negative and positive values. However, only positive items were included in the choice task, and there is a consistently positive effect of attention on choice for other tasks (see Cavanagh et al. 2014; Smith and Krajbich 2018; Smith and Krajbich 2019) with only positive outcomes. This needs to be more openly acknowledged and discussed.

c. The last paragraph of Discussion talks about a lack of benefit of focussed attention in the analysed task. Would focussing attention would become beneficial in decision tasks with more than 2 options? Although answering this question would be a separate paper, a few sentences on generalising this work to more than two options could be included in the Discussion.

5. Coverage of literature.

a. In the Introduction, the authors state that the "final choices are biased towards the item that they looked at longer, irrespective of its desirability". This is not quite true. The desirability does matter, as shown in Smith and Krajbich 2019, as well as Westbrook et al. 2020. Moreover, as the authors note in the discussion, Armel et al. 2008 show that attention has a reverse effect when the items are aversive. Please update the introduction accordingly.

b. The authors do not mention that there is some work that has argued for value attracting attention in multi-alternative choice. While that work does not go to the lengths that this paper does, it does make normative arguments for why this should occur, namely to eliminate non-contenders (Krajbich and Rangel 2011; Towal et al. 2013; Gluth et al. 2020). Finally, the authors might also want to mention Ke, Shen, Villas-Boas (2016), which also takes a normative approach to information search in consumer choice.

c. On page 2 the authors state that "no current normative framework incorporates control of attention as an intrinsic aspect of the decision-making process". This does not seem to be accurate given the study by of Cassey et al. (2013). The key difference is that in Cassey et al. the focus was placed on the fixed duration paradigm, while the present manuscript focuses on the free-response paradigm. Please clarify the link of the current study with Cassey et al.

d. In lines 318-321, the authors state that the model of Cassey et al. "could not predict when they [fixation switches] ought to occur". The model of Cassey et al. does predict when the optimal switching times are, but for the case of a fixed duration paradigm with \kappa = 0. In this case the optimal switch policy is much simpler (single or at most double switch at particular times) than in the free-response paradigm nicely analysed in the present manuscript.

[Editors' note: further revisions were suggested prior to acceptance, as described below.]

Thank you for resubmitting your work entitled "Optimal policy for attention-modulated decisions explains human fixation behavior" for further consideration by *eLife*. Your revised article has been evaluated by Joshua Gold (Senior Editor) and a Reviewing Editor.

Summary:

The authors have done an excellent and thorough job in addressing most of the comments that were raised during the first review. Please find below a list of remaining issues.

The manuscript has been improved but there are some remaining issues that need to be addressed before acceptance, as outlined below:

Essential revisions

1. In our previous points 2a-b the request was to show the switch rate and fixation duration as a function of time, value sum and value difference. We apologise if that was not clear previously but with the term "time" we referred to elapsed time within a trial rather than reaction times (RT). Because RT's will be influenced by various properties of the trial (e.g. trial difficulty) the analyses reported in the revision are not very easy to interpret. Additionally, we are puzzled by the fact that the aDDM model predicts non-flat switch rates and fixation durations as a function of the different quantities (Figure 4—figure supplement 1), given that switching in the aDDM version the authors used is random. The aim of these previous points was to (a) highlight the differences between random switching (aDDM) and deliberate switching (optimal model), and (b) understand how switching tendencies change as a function of trial-relevant quantities and time elapsed in the optimal model. We appreciate that such analyses might be difficult to perform given the interrelationship among time, value sum and absolute value difference. Below we offer a few suggestions:

– One possibility is to consider a "stimulus locked" approach, in which the switch rate and fixation duration is plotted as a function of time elapsed from the stimulus onset and up to "x" milliseconds before the response. Inevitably, later time points will more likely include certain trial types, e.g. only difficult trials (or trials with low value sum). The authors can consider using a "stratification" approach, subsampling the trials such that all time-points have comparable trial distributions (in terms of value difference and value sum). The influence of value difference and value sum can be examined using median splits based on these quantities.

– Specifically for the switch rate analysis, the authors could perform a logistic regression trying to predict at each point in time the probability of switching, also considering covariates such as value sum or absolute value difference.

– Since the last fixation can be cut short, we recommend excluding the last fixation from these analyses.

– We recommend that the scaling of the x and y axis in the data and in the models is the same to allow comparison in absolute terms.

2. Krajbich et al. 2010 used an aDDM implementation in which fixations were not random but instead sampled from the empirical switching times distributions (thus fixations depended on fixation number, trial difficulty etc). The authors currently do not acknowledge this previous implementation. The aDDM with random switching is a good baseline for the scope of this paper, but the version used by Krajbich et al. 2010 makes different predictions than the random aDDM; and this should be explicitly acknowledged. For instance, the aDDM where switching matches the empirical distributions, accounts for the fact that the first fixations were shorter than the rest. Discussing this non-random aDDM used by Krajbich et al. 2010, will fit easily in the current discussion, since the optimal model offers a rationale fort the empirical fixation patterns that the older work simply incorporated into the aDDM simulations.

3. The authors slightly mischaracterise Krajbich and Rangel 2011 by saying that "fixation patterns were assumed to be either independent of the decision-making strategy". That paper did condition fixation patterns on the values of the items. Here is a direct quote from the Discussion: "These patterns are interesting for several reasons. First, they show that the fixation process is not fully independent from the valuation process, and contains an element of choice that needs to be explained in further work."

4. When comparing the mean reward of the models, the authors simulate the aDDM not with the parameters from Krajbich et al. 2010, but with different parameters meant to "ensure a fair comparison" between the models. We believe that this approach sets the aDDM to a disadvantage. If the best-fitting parameters of the "optimal model" lead to a lower signal-to-noise ratio than the aDDM best-fitting parameters, that should be acknowledged and accepted as is. We recommend the authors state upfront that the best-fitting optimal model does not outperform the data or the aDDM. However, if you use a non-best-fitting aDDM, then the aDDM underperforms both.

---

## [Author Response]

Essential revisions1. Link between the normative model and human data.One of the main claims in the paper is that the normative model reproduces key aspects of human behaviour in a value-based binary choice task reported elsewhere (Krajbich et al., 2010). This claim needs to be backed up by additional analyses and clarifications. In particular:a. In Figure 3A, the model predicts a linear choice curve (as opposed to a sigmoidal in the human data) while, also unlike in the data, the RT curves are concave. Are the shapes of these curves a general feature of the model or do they emerge for a certain parametrisation? Would for example the results look different with fixed bounds?

The model predicts linear choice curves in Figure 3A due to the difficulty of the task, which is set by the model parameters including the evidence noise term (𝜎^"^). Therefore, if we set the evidence noise term (𝜎^"^) to a lower value, the model will exhibit sigmoidal choice curves because the decision becomes easier at extreme value differences. Below is the RT curve after decreasing the evidence noise term (𝜎^"^) from 27 to 5, which renders the decisions easier due to an overall more reliable evidence accumulation. This can be seen in Figure 3—figure supplement 1A.

b. Similarly, in Figure 3D, the model shows that the last fixation influences probability of choice equally across value differences. In the human data this effect diminishes as value difference increases. Is this prediction or a feature of the model?

Similar to above, for Figure 3D, decreasing the noise term causes the choice curves to reach an asymptote at less extreme value differences. This is reflected in the last fixation effect, which shows a diminishing effect as the value difference increases. In Figure 3—figure supplement 1B is the last fixation effect curve after decreasing the evidence noise term (𝜎^"^) from 27 to 5.

c. In Figure 3C the authors focus on the number of switches, which however are highly correlated with RTs. Can the authors show in addition the number of switches per unit of time (equivalent to probability of switching) in the human data and in their model?

We thank the reviewers for this suggestion, and agree that the rate of switching is an important measure to report. Human data showed no relationship between rate of switching and trial difficulty (Figure 3—figure supplement 1D; t(38) = -0.32, p = 0.75). Interestingly, we found that with the large number of simulated trials, our optimal model shows an increase in the rate of switching as the task difficulty decreases (Figure 3—figure supplement 1E; t(38) = 2.96, p = 0.0052). However, when using the same number of trials as human data, this relationship was not apparent in the model (Figure 3—figure supplement 1F; t(38) = 1.02, p = 0.31), suggesting the human data may be underpowered to show such a relationship. Furthermore, we find that an increase in switch rate with decreasing trial difficulty is not a general property of the optimal model, as a significant increase in the switch cost (𝐶_$_ from 0.018 to 0.1) reduces the overall number of switches, and removes this effect (Figure 3—figure supplement 1G; t(38) = -0.50, p = 0.62), even with a large number of simulated trials.

We have added the following to the Results to briefly summarize the switch rate analysis:

“Since the number of switches is likely correlated with response time, we also looked at switch rate (number of switches divided by response time). Here, although human data showed no relationship between switch rate and trial difficulty, model behavior showed a positive relationship, suggesting an increased rate of switching for easier trials. However, this effect was absent when using the same number of trials as humans, and did not generalize across all model parameter values (Figure 3—figure supplement 2).”

d. In Figure 4B-C, RTs and "value sum" may covary, with high value sums presumably resulting in shorter RTs. Does the value sum effect persist in the model and in the data once the influence of RTs is accounted for?

To address this, we repeated the analysis for the effect of value sum on the choice bias coefficients using a regression model with both value sum and RT as independent variables. We fit a regression model for each participant, then performed a t-test of the regression coefficients across participants. For simulated data from the model, after accounting for the RT, value sum still had a significant effect on choice bias (t(38) = 7.88, p < 0.001). Conversely, in the regression model for RT, adding value sum as another independent variable still led to a significant effect of RT on the choice bias coefficients (t(38) = -5.73, p < 0.001).

For human data, value sum had a significant effect on choice bias coefficients after adding RT to the regression (t(38) = 2.91, p = 0.006). RT had a non-significant effect on choice bias after adding value sum to the regression, although it was trending in the expected direction (t(38) = -1.32, p = 0.20).

We added the following segment to the Results and Methods and Materials sections:

Results:

“Since response time may be influenced by the sum of the two item values and vice versa, we repeated the above analyses using a regression model that includes both value sum and response time as independent variables (see Methods and Materials). The results were largely consistent for both model (effect of RT on choice bias: t(38) = -5.73, p < 0.001, effect of value sum: t(38) = 7.88, p < 0.001) and human (effect of RT: t(38) = -1.32, p = 0.20, effect of value sum: t(38) = 2.91, p = 0.006) behavior.”

Methods and Materials:

“To test for the effect of RT and value sum on choice bias after accounting for the other variable, we used a similar approach and used both RT and value sum as independent variables in the regression model and the choice bias coefficient as the dependent variable.”

2. Comparison between the normative and aDDM models.The two models differ in the way they conceptualise attentional switching (exogenous vs. endogenous). Does this difference manifest itself in the predictions the two models make? And can the human data distinguish between the two models? The authors comment in the Discussion that their model is more constrained by normative considerations, and may thus fit the data worse than other ad-hoc models. Asides quantitative fitting, which the authors may elect to perform, we would like to see a qualitative comparison between the two models.

Indeed, we had fairly limited qualitative analysis in how our model behavior differs from that of the aDDM, and – potentially – from human data. We have thus performed additional analyses that we describe below. Following this, we provide the text we added to the main text to describe these analyses.

a. How does the probability of switching changes as a function of time (within trial) and as a function of absolute value difference and sum (across trials) in the two models and in the data?

To test the relationship between switch rate and time, for each participant, we divided all trials into five equallysized bins based on RT. The plots show the average curves across participants, where vertical error bars indicate SEM for the relevant y-variable (e.g., switch rate), and horizontal error bars indicate the SEM of the bin means. A linear relationship between the x and y variables were performed by fitting a linear regression model within each participant, then performing a t-test of the regression coefficients across subjects against zero. Analogous binning and statistical procedures were used when dividing trials by value sum and value difference. For model simulations, we used the same trials and their corresponding item values as the human task.

In human data, the probability of switching decreases as a function of time (t(38) = -4.49, p < 0.001), while this relationship is neither apparent in the optimal model nor the aDDM (Author response image 1).

However, the shape of the optimal model curve suggested that trials with only a single fixation (switch rate = 0) may be distorting the relationship between switch rate and time. When only including trials where at least one switch occurred, both models predicted a decrease in switch rate over time, consistent with human data (optimal model: t(38) = -29.6, p < 0.001, aDDM: t(38) = -7.70, p < 0.001). This suggests that in both models, single fixation trials significantly affect the switch rate (Author response image 2).

**Author response image 2. respfig2:** 

The shape of the curve before and after removing single fixation trials suggests that the RT distribution in single fixation trials are more tightly distributed in the optimal model compared to the aDDM. Plotting the RT distribution of these trials confirmed this prediction.

**Author response image 3. respfig3:** 

Regarding the relationship between switch rate and value sum, human data showed no significant relationship (value sum, t(38) = -0.84, p = 0.40). However, both the optimal model and the aDDM showed a negative association, such that switch rate decreased as the value sum increased, suggesting that the model is less likely to switch attention within the same time frame for trials where higher value items are being compared (optimal model, t(38) = -4.11, p < 0.001; aDDM, t(38) = -2.09, p = 0.044) (Author response image 4).

**Author response image 4. respfig4:** 

Regarding the relationship between switch rate and absolute value difference (i.e., trial difficulty), human data again showed no significant relationship (t(38) = -0.67, p = 0.51). The optimal model also showed no significant relationship between switch rate and value difference (t(38) = -0.41, p = 0.68). However, the aDDM showed a positive association, suggesting that more switches occurred within the same time-frame for easier trials (t(38) = 4.62, p < 0.001) (Author response image 5).

**Author response image 5. respfig5:** 

To summarize, human data and both models show a decrease in switch rate as a function of time, suggesting that while the number of switches increases as a function of time, optimal behavior involves decreasing the likelihood of switching attention within the same time frame as the deliberation process takes longer. In contrast, we did not see a consistent relationship between switch rate and value sum or value difference for both human and model behavior. Of note, both the optimal model and aDDM show similar patterns when using a large number of simulated trials rather than the same trials used in human data, such that both models predict a negative association between switch rate and value sum, and a positive association between switch rate and absolute value difference. Therefore, we suspect that human data may still exhibit a similar relationship if more trials are performed. We report these findings in Figure 4—figure supplement 1A-D and its corresponding figure legend.

b. How does the fixation duration changes as a function of time (within trial) and as a function of absolute value difference and sum (across trials) in the two models and in the data?

Human data shows a positive association between fixation duration and RT (t(38) = 9.28, p < 0.001), a negative association between fixation duration and value sum (t(38) = -2.81, p = 0.0078), and a negative association between fixation duration and value difference (t(38) = -5.46, p < 0.001). Both the optimal model and the aDDM predicted similar patterns (optimal model, RT: t(38) = 85.6, p < 0.001 , value sum: t(38) = -4.19, p < 0.001, value diff: t(38) = -3.60, p < 0.001; aDDM, RT: t(38) = 13.65, p < 0.001, value sum: t(38) = -3.32, p = 0.002, value diff: t(38) = -6.44, p < 0.001) (Author response image 6).

**Author response image 6. respfig6:** 

c. The model seems to predict a non-negligible number of single fixation trials. How does this align with the data and the aDDM predictions?

While both the optimal model and aDDM over-estimated the number of single fixation trials compared to human data, the aDDM predicted significantly more than the optimal model (t(76) = 5.84, p < 0.001) (Author response image 7).

**Author response image 7. respfig7:** 

d. How do reaction time distributions look like under the normative model? Are these comparable to the RT distributions in the data?

Both models predict a RT curve that includes more sub-1s trials, consistent with the above results showing more single-fixation trials (Figure 4—figure supplement 3F).

e. Please also show the aDDM predictions together with the novel predictions made by the normative model in Figure 4C.

The aDDM also replicated the effects of RT and value sum on fixation bias (RT: t(38) = -48.6, p < 0.001; value sum: t(38) = 14.7, p < 0.001). This is not surprising given the initial assumptions made by the model wherein fixations boost the value of the item. Since increasing RT allows the model to spend similar amounts of time to each item, fixation bias will decrease. Also, since the impact of fixation is proportional to the value of the items, choosing between items with higher value will lead to a stronger fixation bias (Figure 4—figure supplement 3C and D).

f. The comparison between the two models is currently done on the basis of mean reward. How do the predictions of the models compare to the mean reward accrued by human participants in Krajbich et al? Please also clarify in the main text that the models are not compared on the basis of goodness of fit. In particular, in the paragraph starting in line 260, terms such as "outperformed", "competitive performance", "comparison" are ambiguous.

To test for this, we performed an independent samples t-test to the mean reward achieved by human participants versus the simulated participants of both models. We found there is no significant difference between the mean rewards of humans versus the optimal model (t(76) = 0.69, p = 0.49) and humans versus the aDDM (t(76) = -0.062, p = 0.95). Of note, for this analysis we used the aDDM setup used in the original paper by Krajbich et al., 2010 rather than the signal-to-noise-matched version we used to compare the mean reward between the optimal model and aDDM. Therefore, the mean reward of the optimal model was not greater than that of the aDDM in this scenario. To calculate mean reward, we used the same cost per unit time used for the optimal model (c = 0.23) (Figure 4—figure supplement 3B).

We also modified the ambiguous terms the reviewer mentioned:

“We also tested to which degree the optimal model yielded a higher mean reward than aDDM, which, despite its simpler structure, could nonetheless collect competitive amounts of reward. To ensure a fair comparison, we adjusted the aDDM model parameters (i.e., attentional value discounting and the noise variance) so that the momentary evidence provided to the two models has equivalent signal-to-noise ratios (see Supplementary file 1).”

g. Please explain why the model shows the pattern that is demonstrated in Figure 3F. Is this pattern also predicted by the aDDM?

The aDDM did not predict the same fixation pattern as the data and optimal model. This fixation pattern in the optimal model is well-preserved across different parameter values. We suspect this pattern arises due to the shape of the optimal decision boundaries, where the particle is more likely to hit the “switch” boundary in a shorter time for the first fixation, since the model likely prefers to sample from both items at least once. Consistent with this, Figure 2C shows that the “accumulate” space is larger for the second fixation compared to the first (Author response image 8).

**Author response image 8. respfig8:** 

We added the following to the Results:“Interestingly, the model also replicated a particular fixation pattern seen in humans, where a short first fixation is followed by a significantly longer second fixation, which is followed by a medium-length third fixation (Figure 3F). We suspect this pattern arises due to the shape of the optimal decision boundaries, where the particle is more likely to hit the “switch” boundary in a shorter time for the first fixation, likely reflecting the fact that the model prefers to sample from both items at least once. Consistent with this, Figure 2C shows that the “accumulate” space is larger for the second fixation compared to the first fixation. Of note, the attentional drift diffusion model (aDDM) that was initially proposed to explain the observed human data (Krajbich et al., 2010) did not show this fixation pattern Figure 4—figure supplement 2D.”

Overall, we added the following Figures / panels to describe the comparison to aDDM: Figure 3—figure supplement 2G, Figure 4—figure supplement 1, and Figure 4—figure supplement 2. Furthermore, we added the following to the main text:

“Next, we assessed how the behavioral predictions arising from the optimal model differed from those of the original attentional drift diffusion model (aDDM) proposed by Krajbich et al., (2010). Unlike our model, the aDDM follows from traditional diffusion models rather than Bayesian models. It assumes that inattention to an item diminishes its value magnitude rather than the noisiness of evidence accumulation. Despite this difference, the aDDM produced qualitatively similar behavioral predictions as the optimal model (Figure 3—figure supplement 2G, Figure 4—figure supplement 1), although the optimal model was able to better reproduce some of the fixation patterns seen in human behavior (Figure 4—figure supplement 2A,D).”

3. Comprehensibility of the modelling.The authors have done an admirable job of boiling down some heavy mathematics (in the SI) into just the key steps in the main text. A lot of this builds on prior work in Tajima et al. Still, the authors could do more to explain the math, which would make this paper more self-contained and really help the reader understand the core ideas/intuitions.a. Equation 1 – it isn't obvious how the mean will behave over time or where this expressions comes from. It would be helpful to state that the z part is the prior, and that as t->infinity the σ terms become negligible and the expression converges to x/t, namely the true value. The σ-squared terms seem to come out of nowhere. In the methods the authors explain that this has to do with Fisher information, but most readers won't know what that is or why it is the appropriate thing to include here. Also, sigma_x_ is defined later in the text, but should be defined up front here.

We agree that this expression might come as a surprise to some. We could have simplified it slightly, with the downside of a less direct relation to Eq. (2). As we felt that establishing this relationship is essential, we decided to keep Eq. (1) in its more complex form. To nonetheless make Eq. (1) it easier to digest, we have added additional information about how the prior and likelihood variances relate to their respective informativeness. Additionally, we have added additional details about the structure of the posterior mean and how it changes with increasing accumulation time t.

In particular, we added that “the smaller the prior variance (𝜎_%_^"^), the more information this prior provides about the true values”, and that “the evidence accumulation variance (𝜎_and_^"^) controls how informative the momentary evidence is about the associated true value. A large 𝜎_and_^"^ implies larger noise, and therefore less information provided by each of the momentary evidence samples.” We were also more explicit in describing how the posterior mean and variance evolves over time: “The mean of this posterior (i.e., the first fraction in brackets) is a weighted sum of the prior mean, 𝑧, and the accumulated evidence, 𝑥(𝑡). The weights are determined by accumulation time, 𝑡, and the variances of the prior, 𝜎_%_^"^, and the momentary evidence, 𝜎_and_^"^, which control their respective informativeness. Initially, 𝑡 = 0 and 𝑥(𝑡) = 0, such that the posterior mean equals that of the prior, 𝑧. Over time, with increasing 𝑡, the influence of 𝑥(𝑡) = 0 becomes dominant, and the mean approaches 𝑥(𝑡)/𝑡 (i.e., the average momentary evidence) for large 𝑡, at which point the influence of the prior becomes negligible. The posterior's variance (i.e., the second fraction in brackets) reflects the uncertainty in the decision maker's value inference. It initially equals the prior variance, 𝜎_%_^"^, and drops towards zero once 𝑡 becomes large.”

We refrained from mentioning ‘Fisher information’ in the main text to avoid confusing readers that are not familiar with this concept. Instead, we kept discussion at a more informal level. Furthermore, we have added additional information about why we assess informativeness by Fisher information to Methods and Materials:

**“**We measure how informative a single momentary evidence sample is about the associated true value by computing the Fisher information it provides about this value. This Fisher information sums across independent pieces of information. This makes it an adequate measure for assessing the informativeness of momentary evidence, which we assume to be independent across time and items.**”**

b. Between Equation 1 and 2: In the means for delta_x_, one has a delta_t_ multiplying z and one doesn't. Why? Is this a typo? The delta_t_ should always be there, but again, it isn't obvious.c. Equation 2: Why does the mean have a z1| at the beginning of it? Is that a typo?

Both instances are indeed typos, and we thank the reviewers for pointing them out. We have corrected all instances of such typos from the main text and the Supplementary file 1.

d. Could the authors elaborate more on why/when the decision-maker chooses to switch attention? They say that the decision at each time point only depends on the difference in posterior means but Figure 2c seems to indicate that if the difference in posterior means stays constant over a period of time, then the process enters the "switch" zone and shifts attention.

Optimal decisions are determined as a function of not only the difference in posterior means (𝛥), but also the times attended to item 1 (𝑡_1_) and item 2 (𝑡_"_). This results in an optimal policy shape where the particle will hit the “switch” zone if the difference in expected rewards between the two items is too small to make an immediate decision, and it is deemed advantageous to collect more reward-related evidence of the currently unattended item. This prevents the model from deliberating for too long while attending to a single item. We added this clarification in Results under ‘Features of the Optimal Policy.’:

“In other words, the difference in expected rewards between the two items is too small to make an immediate decision, and it is deemed advantageous to collect more information about the currently unattended item.”

4. Model assumptions.a. The model assumes that attention can take one of two discrete states. However, attention is often regarded as operating in a more graded fashion, which would necessitate parameter kappa in the model to be free to change within a trial. This will probably render the model intractable. However, a more viable and less radical assumption, would be to allow for a third "divided attention" mode in which the agent samples equally from both alternatives (divided attention mode). If this extension is technically challenging, one conceptual question that the authors can discuss in the paper, is whether the normative model would ever switch away from this divided attention mode.

We agree that a decision-making model that incorporates a continuously variable attention would be an interesting endeavor. As the reviewers suggest, we could address this by adding another state to the Bellman equation in which attention is perfectly divided. Since this will add another dimension to the value and policy space, we anticipate this will become intractable quickly. However, we believe there is sufficient literature on this topic to reasonably predict how such a model would behave.

Previous work by Fudenberg, Strack and Strzalecki (2018) discusses a modified drift diffusion model in which attention can vary continuously and gradually across two choice options. They show that, consistent with our results, the model with equally divided attention performs optimally. Drawing from this, we can confidently state that our optimal model would always engage in the divided attention mode. However, the authors also state that there may be instances within a decision when it would be optimal to pay unequal attention. In fact, if the normative decision maker has already paid more attention to one item over the other item, it may be optimal to switch attention and gain more information about the unattended item rather than to proceed in the divided attention mode.

To address this, we added the following to the Discussion:

“We show that narrowing the attentional bottleneck by setting κ to values closer to 0 or 1 does not boost performance of our decision-making model (Figure 4E). Instead, spreading a fixed cognitive reserve evenly between the attended and unattended items maximized performance. This is consistent with prior work that showed that a modified drift diffusion model with a continuously varying attention would perform optimally when attention is always equally divided (Fudenberg et al., 2018). However, this does not necessarily imply that equally divided attention always constitutes the normative behavior. If the decision maker has already paid more attention to one item over the other within a decision, it may be optimal to switch attention and gain more information about the unattended item rather than to proceed with equally divided attention.”

b. The authors need to assume a certain prior, namely z_bar = 0, in order to always get a positive effect of attention. This seems like an important controversy in the model; it is a noticeably non-Bayesian feature of a Bayesian model. The authors try to explain this away by noting that the original rating scale included both negative and positive values. However, only positive items were included in the choice task, and there is a consistently positive effect of attention on choice for other tasks (see Cavanagh et al. 2014; Smith and Krajbich 2018; Smith and Krajbich 2019) with only positive outcomes. This needs to be more openly acknowledged and discussed.

We agree with the reviewers that the exact features and role of the Bayesian prior remains a topic of discussion, and we acknowledge that while our formulation suggests a zero-mean prior, there is also evidence suggesting the prior should be centered on the choice set. We have modified the Discussion to be more transparent regarding this point, and added the citations suggested by the reviewers in support of a non-zeromean prior distribution.

“In our model, we assumed the decision maker's prior belief about the item values is centered at zero. In contrast, Callaway et al. (2020) chose a prior distribution based on the choice set, centered on the average value of only the tested items. While this is also a reasonable assumption (Shenhav et al., 2018), it likely contributed to their inability to demonstrate the choice bias. Under the assumption of our zero-mean prior, formulating the choice process through Bayesian inference revealed a simple and intuitive explanation for choice biases (Figure 4A) (see also Li and Ma (2020)). This explanation required the decision maker to a-priori believe the items' values to be lower than they actually are when choosing between appetitive options, consistent with evidence that item valuations vary inversely with the average value of recently observed items (Khaw et al., 2017). The zero-mean prior also predicts an opposite effect of the choice bias when deciding between aversive items, such that less-fixated items should become the preferred choice. This is exactly what has been observed in human decision-makers (Armel et al., 2008). We justified using a zero-mean bias because participants in the decision task were allowed to rate items as having both positive or negative valence (negative-valence items were excluded from the binary decision task). However, there is some evidence that humans also exhibit choice biases when only choosing between appetitive items (Cavanagh et al., 2014, Smith et al., 2018, Smith et al., 2019). Although our setup suggests a zero-mean prior is required to reproduce the choice bias, the exact features and role of the Bayesian prior in human decisions still remains an open question for future work.”

c. The last paragraph of Discussion talks about a lack of benefit of focussed attention in the analysed task. Would focussing attention would become beneficial in decision tasks with more than 2 options? Although answering this question would be a separate paper, a few sentences on generalising this work to more than two options could be included in the Discussion.

We agree it is interesting to discuss whether our findings would generalize to multi alternative choices. While we cannot definitively answer this question, we believe that under the same framework of our binary choice model, the same principles would apply such that divided attention across all items would lead to optimal behavior. Once the framework becomes more complex and involves features such as increased attention to items based on value or salience, this may lead to scenarios where focused attention may be beneficial. This is consistent with the idea that although divided attention maximizes reward on average, focusing attention to single items may be preferred if the decision maker has already done so for any other item(s) for heuristical reasons.

We have added the following text to the Discussion to highlight this point:

“An open question is whether our findings can be generalized to multi-alternative choice paradigms (Towal et al., 2013, Ke et al., 2016, Gluth et al., 2020, Tajima et al., 2019). While implementing the optimal policy for such choices may be analytically intractable, we can reasonably infer that a choice bias driven by a zero-mean prior would generalize to decisions involving more than two options. However, in a multi alternative choice paradigm where heuristics involving value and salience of items may influence attention allocation, it is less clear whether an equally divided attention among all options would still maximize reward. We hope this will motivate future studies that investigate the role of attention in more realistic decision scenarios.”

5. Coverage of literature.a. In the Introduction, the authors state that the "final choices are biased towards the item that they looked at longer, irrespective of its desirability". This is not quite true. The desirability does matter, as shown in Smith and Krajbich 2019, as well as Westbrook et al. 2020. Moreover, as the authors note in the discussion, Armel et al. 2008 show that attention has a reverse effect when the items are aversive. Please update the introduction accordingly.

Thank you for this suggestion. We have now removed the phrase “irrespective of its desirability.”

b. The authors do not mention that there is some work that has argued for value attracting attention in multi-alternative choice. While that work does not go to the lengths that this paper does, it does make normative arguments for why this should occur, namely to eliminate non-contenders (Krajbich and Rangel 2011; Towal et al. 2013; Gluth et al. 2020). Finally, the authors might also want to mention Ke, Shen, Villas-Boas (2016), which also takes a normative approach to information search in consumer choice.

We agree that the effect of value and salience of the choice items on attention allocation is a relevant topic to discuss. We also agree that these processes are likely contributing to the fixation behavior of human participants, and elements that could be added to our normative formulation in future work. We have added the following text to note this, and added the citations, including Ke et al., (2016), suggested by the reviewer (see bolded text for added information).

“In previous work, fixation patterns were assumed to be either independent of the decision-making strategy (Krajbich et al., 2010, Krajbich et al., 2011) or generated by heuristics that relied on features such as the salience or value estimates of the choice options (Towal et al., 2013, Gluth et al., 2020). Other models generated fixations under the assumption that fixation time to different information sources should depend on the expected utility or informativeness of the choice items (Ke et al., 2016, Cassey et al., 2013, Song et al., 2019).”

“When designing our model, we took the simplest possible approach to introduce an attentional bottleneck into normative models of decision-making. When doing so, our aim was to provide a precise (i.e., without approximations), normative explanation for how fixation changes qualitatively interact with human decisions rather than quantitatively capture all details of human behavior, which is likely driven by additional heuristics and features beyond the scope of our model (Acerbi et al., 2014, Drugowitsch et al., 2016). For instance, it has been suggested that normative allocation of attention should also depend on the item values to eliminate non-contenders, which we did not incorporate as a part of our model (Towal et al., 2013, Gluth et al., 2020). As such, we expect other models using approximations to have a better quantitative fit to human data (Krajbich et al., 2010, Callaway et al., 2020).”

“An open question is whether our findings can be generalized to multi-alternative choice paradigms (Towal et al., 2013, Ke et al., 2016, Gluth et al., 2020, Tajima et al., 2019).”

c. On page 2 the authors state that "no current normative framework incorporates control of attention as an intrinsic aspect of the decision-making process". This does not seem to be accurate given the study by of Cassey et al. (2013). The key difference is that in Cassey et al. the focus was placed on the fixed duration paradigm, while the present manuscript focuses on the free-response paradigm. Please clarify the link of the current study with Cassey et al.

We agree with this point, and modified the statement to the following:

“While several prior studies have developed decision-making models that incorporate attention (Yu et al., 2009, Krajbich et al., 2010,Towal et al., 2013, Cassey et al., 2013, Gluth et al., 2020), our goal was to develop a normative framework that incorporates control of attention as an intrinsic aspect of the decision-making process in which the agent must efficiently gather information from all items while minimizing the deliberation time, akin to real life decisions. In doing so, we hoped to provide a computational rationale for why fixationdriven choice biases seen in human behavior may arise from an optimal decision strategy.”

d. In lines 318-321, the authors state that the model of Cassey et al. "could not predict when they [fixation switches] ought to occur". The model of Cassey et al. does predict when the optimal switching times are, but for the case of a fixed duration paradigm with \kappa = 0. In this case the optimal switch policy is much simpler (single or at most double switch at particular times) than in the free-response paradigm nicely analysed in the present manuscript.

We thank the reviewers for this clarification, and modified the statement to the following:

“Furthermore, since their decision task involved a fixed-duration, attention switches also occurred at fixed times rather than being dynamically adjusted across time, as in our case with a free-response paradigm.”

[Editors' note: further revisions were suggested prior to acceptance, as described below.]

Essential revisions1. In our previous points 2a-b the request was to show the switch rate and fixation duration as a function of time, value sum and value difference. We apologise if that was not clear previously but with the term "time" we referred to elapsed time within a trial rather than reaction times (RT). Because RT's will be influenced by various properties of the trial (e.g. trial difficulty) the analyses reported in the revision are not very easy to interpret. Additionally, we are puzzled by the fact that the aDDM model predicts non-flat switch rates and fixation durations as a function of the different quantities (Figure 4—figure supplement 1), given that switching in the aDDM version the authors used is random. The aim of these previous points was to (a) highlight the differences between random switching (aDDM) and deliberate switching (optimal model), and (b) understand how switching tendencies change as a function of trial-relevant quantities and time elapsed in the optimal model. We appreciate that such analyses might be difficult to perform given the interrelationship among time, value sum and absolute value difference. Below we offer a few suggestions:– One possibility is to consider a "stimulus locked" approach, in which the switch rate and fixation duration is plotted as a function of time elapsed from the stimulus onset and up to "x" milliseconds before the response. Inevitably, later time points will more likely include certain trial types, e.g. only difficult trials (or trials with low value sum). The authors can consider using a "stratification" approach, subsampling the trials such that all time-points have comparable trial distributions (in terms of value difference and value sum). The influence of value difference and value sum can be examined using median splits based on these quantities.– Specifically for the switch rate analysis, the authors could perform a logistic regression trying to predict at each point in time the probability of switching, also considering covariates such as value sum or absolute value difference.– Since the last fixation can be cut short, we recommend excluding the last fixation from these analyses.– We recommend that the scaling of the x and y axis in the data and in the models is the same to allow comparison in absolute terms.

Thank you for the clarification, and we apologize for the misunderstanding.

We reanalyzed the data completely in order to address the discrepancies the editors have brought up. For the aDDM simulations in the previous version of the manuscript, we already used the empirical distribution of fixation durations as described in the original literature (Krajbich et al., 2010), but did not sample the first fixations separately. As we discuss below (point #2), we do so now. We also noted that in the human data, RT was not equal to the sum of all fixation durations since there were procedures to remove non-item fixations by the original authors. This likely added to some discrepancy between the aDDM simulations and human data, especially for the switch rate analysis where we divided the number of switches by the RT. For the following analyses, we have adjusted the behavioral data so that RT is now equal to the sum of all fixations.

First, we would like to address the reviewers’ point that the aDDM model should predict flat switch rates and fixation duration as a function of different quantities. As the reviewers correctly point out, the aDDM samples fixation durations from human data, separately for each absolute value difference between the two items. Therefore, the relationship between switch probability and other variables such as time, value difference, and value sum should be reasonably preserved between the model and human data. However, we find that the implementation of the aDDM results in some systematic discrepancies in the switching behavior. For instance, since the aDDM terminates the trial whenever a decision boundary is reached, long fixations are more likely to result in boundary crossings than short fixations, such that long fixations are more likely considered last fixations and thus excluded from analysis. This causes the model to feature lower mean middle fixation durations and a higher switch frequency when compared to human data.

In Author response image 9, we show how middle fixation duration and switch rate varies across trials based on the trial RT, absolute value difference, and value sum for humans and the aDDM simulations. For every trial, we computed the mean middle fixation duration and switch rate (total number of switches divided by RT), then grouped the trials based on RT (using 0.5s bins from 1-4s), absolute value difference (i.e., trial difficulty) and value sum. We then computed the mean fixation duration and switch rate across participants for each x-variable. We show these results mainly to demonstrate that, although the aDDM randomly samples non-first fixations from empirical middle fixation durations, trial variables such as RT and item values still influence the switching behavior when implemented in the aDDM framework.

**Author response image 9. respfig9:** 

The plots are shown in Author response image 9, along with analogous plots for the optimal model, for the new Figure 4—figure supplement 1. The purpose of this figure is to show a clearer version of our between-trial analyses of how switch rate and fixation duration are affected by value sum and value difference for the human and the two simulated behavioral data sets. We did not show the effect of RT on these quantities since, as the reviewers mention, this is difficult to interpret since RT correlates with other task variables such as difficulty. We also decided to use different y-axis scales, since using the same scale for all three datasets makes it difficult to appreciate the slope of certain plots.We next turned to the original intent of the review question, which is to show how switch probability and fixation duration varies across time within trials. To do so, we followed the reviewers’ recommended analyses. For a given participant, we aligned all trials by the stimulus onset, then counted the number of switches within each 200ms time bin. We then averaged the switch count within each time bin across trials to get the switch probability. Since RTs differ across trials, we only included time points up to when at least ⅓ of the total trials are included. This implies that switch probabilities at later times are averaged across fewer trials. We also removed the last fixations as suggested.

We found that switching behavior within trials is well-preserved between humans and the aDDM. Both humans and the aDDM show a peak in switch probability within 1s of stimulus onset, followed by a gradual decrease (Figure 4—figure supplement 2 column A). The optimal model, in comparison, exhibits more discrete time points where a majority of attention switches occur. This is not surprising given the shape of the optimal policy space, where the particle is guaranteed to hit either a switch or decision boundary within a fixed time period for each fixation. Note, however, that these simulations assumed that time is perceived and measured with millisecond precision by the decision-maker, while humans are known to feature noisy time estimates (Buhusi and Meck, 2005). Furthermore, we assumed the additive non-decision time to be noise-free. We therefore anticipate that, with noisy time perception, the shape of the curve will smoothen out and approach that of human behavior and the aDDM.

Next, we explored whether the relationship between switch probability and time is influenced by variables such as RT (Figure 4—figure supplement 2 column B), value sum (Figure 4—figure supplement 2 column C) and value difference (Figure 4—figure supplement 2 column D). To do so, for each participant, we split all trials into three equally sized bins based on the variable of interest, and then made the same plots as column A only including the first and last bin. At each time point, we performed a t-test across participants between the two bins, and marked any time point with a significant difference across bins (Bonferroni corrected) with an asterisk. Humans and aDDM simulations featured a higher switch probability across time for trials with longer RTs compared to trials with shorter RTs, likely reflecting the fact that trials with longer RTs are more difficult, resulting in more early attention switches. Consistent with this, both human data and aDDM simulations showed a slightly higher switch probability across time for trials with low value difference (i.e., difficult trials) compared to those with a high value difference (i.e., easy trials). Within the time periods in which switches occurred, the optimal model featured comparable patterns.

We used a similar method to investigate fixation duration over time. Whenever a switch occurred within a trial, we recorded the fixation duration until the next switch occurred. We only used middle fixations (excluding first and last fixations), similar to the analyses performed in Krajbich et al. (2010). We averaged the fixation durations at each time point across trials, dropping all time points that contain data from less than ⅓ of all trials. We then plotted the mean fixation duration at each time point across participants, shown in Author response image 10.

**Author response image 10. respfig10:** 

The results show that under the current parameterization, the optimal model features overall longer fixation durations than humans. Interestingly, these durations increased for both humans and the optimal model with time, suggesting that more time is allotted to each fixation as the trial becomes more difficult (human: t(38)=4.50, p < 0.001; optimal model: t(38)=46.4, p<0.001). This trend is not seen in the aDDM which draws all middle fixations randomly from the same empirical distribution (t(38)=-0.57, p=0.57). Therefore, any effect of time within a single trial on fixation duration would be eliminated.

2. Krajbich et al. 2010 used an aDDM implementation in which fixations were not random but instead sampled from the empirical switching times distributions (thus fixations depended on fixation number, trial difficulty etc). The authors currently do not acknowledge this previous implementation. The aDDM with random switching is a good baseline for the scope of this paper, but the version used by Krajbich et al. 2010 makes different predictions than the random aDDM; and this should be explicitly acknowledged. For instance, the aDDM where switching matches the empirical distributions, accounts for the fact that the first fixations were shorter than the rest. Discussing this non-random aDDM used by Krajbich et al. 2010, will fit easily in the current discussion, since the optimal model offers a rationale fort the empirical fixation patterns that the older work simply incorporated into the aDDM simulations.

We thank the reviewers for this suggestion, as it brought to attention an oversight in our aDDM simulations. In our previous analyses, we simulated the aDDM by sampling all fixation durations from middle fixations in human data (split by absolute value difference), but did not sample the first fixation separately from the other fixations. This led to the result on the previous Figure 4—figure supplement 2D, where the fixation duration of the first three fixations for the aDDM did not replicate the patterns seen in humans. We have now repeated the same analyses while separately sampling the first fixations which leads to an aDDM prediction that better mimics human fixation behavior. We added this new result to Figure 4—figure supplement 3E.

However, as the reviewers correctly point out, a strength of our model is that it provides a rationale for the fixation patterns seen in humans, whereas the aDDM simply sampled from the empirical distribution. We have now removed all discussions stating that the optimal model better-predicted this fixation pattern compared to the aDDM and clarified this point in the Results section as follows:

“Of note, the attentional drift diffusion model (aDDM) that was initially proposed to explain the observed human data (Krajbich et al., 2010) did not generate its own fixations, but rather used fixations sampled from the empirical distribution of human subjects. Furthermore, they were only able to achieve this fixation pattern by sampling the first fixation, which was generally shorter than the rest, separately from the remaining fixation durations.”

3. The authors slightly mischaracterise Krajbich and Rangel 2011 by saying that "fixation patterns were assumed to be either independent of the decision-making strategy". That paper did condition fixation patterns on the values of the items. Here is a direct quote from the Discussion: "These patterns are interesting for several reasons. First, they show that the fixation process is not fully independent from the valuation process, and contains an element of choice that needs to be explained in further work."

We thank the reviewers for pointing this out, and agree that the Krajbich et al. (2010) as well as the Krajbich and Rangel (2011) papers suggest that fixations are not completely independent of the decision-making strategy, but rather, that they are affected by decision-relevant variables such as trial difficulty or item value. We have modified our Discussion to reflect this:

“This is consistent with previous work that showed that fixation patterns were influenced by variables relevant for the decision, such as trial difficulty or the value of each choice item (Krajbich et al., 2010; Krajbich and Rangel, 2011). However, prior models of such decisions assumed an exogenous source of fixations (Krajbich et al., 2010; Krajbich and Rangel, 2011) or generated fixations using heuristics that relied on features such as the salience or value estimates of the choice options (Towal et al., 2013; Gluth et al., 2020).”

4. When comparing the mean reward of the models, the authors simulate the aDDM not with the parameters from Krajbich et al. 2010, but with different parameters meant to "ensure a fair comparison" between the models. We believe that this approach sets the aDDM to a disadvantage. If the best-fitting parameters of the "optimal model" lead to a lower signal-to-noise ratio than the aDDM best-fitting parameters, that should be acknowledged and accepted as is. We recommend the authors state upfront that the best-fitting optimal model does not outperform the data or the aDDM. However, if you use a non-best-fitting aDDM, then the aDDM underperforms both.

We understand the reviewers’ concern here, and believe that the method in which we perform the comparison depends on the question we are trying to answer. Our goal was to establish that, since our model provides the optimal solution to the decision problem under the current assumptions, it should outperform or at least match the performance of any alternative model. Therefore, it is an attempt to demonstrate optimality of our model, rather than necessarily make any conclusions about the performance of the aDDM or any other model we may use as a comparison. However, the editors correctly point out that another relevant question would be to compare the performance of the best-fitting versions of each model. To address both perspectives, we have now provided the results to both instances, and changed the text in the Results section as following:

“Given that our model provides the optimal solution to the decision problem under the current assumptions, it is expected to outperform, or at least match, the performance of alternative models. To ensure a fair comparison, we adjusted the aDDM model parameters (i.e., attentional value discounting and the noise variance) so that the momentary evidence provided to the two models has equivalent signal-to-noise ratios (see Appendix 1). Using the same parameters fit to human behavior without this adjustment in signal-to-noise ratio yielded a higher mean reward for the aDDM model (t(76) = -14.8, p < 0.001), since the aDDM receives more value information at each time point than the optimal model. The original aDDM model fixed the decision boundaries at $\pm1$ and subsequently fit model parameters to match behavioral data. Since we were interested in comparing mean reward, we simulated model behavior using incrementally increasing decision barrier heights, looking for the height that yields the maximum mean reward (Figure 4D). We found that even for the best-performing decision barrier height, the signal-to-noise ratio-matched aDDM model yielded a significantly lower mean reward compared to that of the optimal model (t(76) = 3.01, p = 0.0027).”